# Genetic screening identifies a SUMO protease dynamically maintaining centromeric chromatin

Sreyoshi Mitra[1,2], Dani L. Bodor[2,3], Ana F. David[2,5], Izma Abdul-Zani[1], João F. Mata[2], Beate Neumann[4], Sabine Reither[4], Christian Tischer[4] & Lars E.T. Jansen [1,2]*

Centromeres are defined by a self-propagating chromatin structure based on stable inheritance of CENP-A containing nucleosomes. Here, we present a genetic screen coupled to pulse-chase labeling that allow us to identify proteins selectively involved in deposition of nascent CENP-A or in long-term transmission of chromatin-bound CENP-A. These include factors with known roles in DNA replication, repair, chromatin modification, and transcription, revealing a broad set of chromatin regulators that impact on CENP-A dynamics. We further identify the SUMO-protease SENP6 as a key factor, not only controlling CENP-A stability but virtually the entire centromere and kinetochore. Loss of SENP6 results in hyper-SUMOylation of CENP-C and CENP-I but not CENP-A itself. SENP6 activity is required throughout the cell cycle, suggesting that a dynamic SUMO cycle underlies a continuous surveillance of the centromere complex that in turn ensures stable transmission of CENP-A chromatin.

[1] Department of Biochemistry, University of Oxford, South Parks Road, Oxford OX1 3QU, UK. [2] Instituto Gulbenkian de Ciência, 2780-156 Oeiras, Portugal. [3] MRC-Laboratory for Molecular Cell Biology, UCL, London WC1E 6BT, UK. [4] European Molecular Biology Laboratory (EMBL), Meyerhofstrasse 1, D-69117 Heidelberg, Germany. [5] Present address: Institute of Molecular Biotechnology, Dr. Bohr-Gasse 3, 1030 Vienna, Austria. *email: lars.jansen@bioch.ox.ac.uk

Human centromeres are defined by an unusual chromatin domain that features nucleosomes containing the H3 variant CENP-A[1–4]. While these nucleosomes typically assemble on α-satellite sequences, they undergo a chromatin-based self-templated duplication along the cell cycle that is largely independent from local DNA sequence features[5–7]. Indeed, CENP-A is sufficient to initiate a centromere and render it heritable through the mitotic cell cycle[6,7]. The maintenance of centromeric chromatin depends on (1) stable transmission of CENP-A nucleosomes across multiple cell cycles[8–10], (2) a template directed assembly mechanism that depends on previously incorporated CENP-A[5,11–15], and (3) cell cycle regulated inheritance and assembly to ensure centromeric chromatin is replicated only once per cell cycle[10,14,16–20].

Several factors involved in the assembly of CENP-A histones into centromeric nucleosomes have been identified[5,21–26]. However, comparatively little is known about how CENP-A, once assembled into chromatin, is stably transmitted from one cell cycle to the next. This is relevant as we previously found CENP-A to have a longer half-life than the canonical histone counter parts[8]. Photoactivation experiments have shown that this unusually high degree of stability is restricted to centromeres[9], a finding recently borne out in genome-wide Chromatin Immunoprecipitation (ChIP) analysis[27]. This suggests that the degree of CENP-A retention in chromatin is a regulated process that is dependent on the centromere. Indeed, an initial insight came from biophysical measurements of the CENP-A nucleosome in complex with its direct binding partner CENP-C that stabilizes CENP-A nucleosomes both in vitro and in vivo[9,28,29] although it is not known whether this occurs at a specific transaction along the cell cycle.

We have previously employed SNAP tag-based fluorescent pulse-chase labeling to visualize the turnover of chromatin-bound CENP-A molecules across multiple cell divisions[8,30]. In addition, using an adapted quench-chase-pulse labeling technique we have used SNAP to track the fate of nascent CENP-A[8,10,19,20,30]. As stable transmission of CENP-A nucleosomes appears central to the epigenetic maintenance of centromeres we sought to identify factors that specifically control CENP-A stability in chromatin.

Here, we present a dedicated assay that allows for the discovery of factors specifically involved in either CENP-A maintenance, CENP-A assembly or both. We construct a custom siRNA library targeting genes encoding chromatin associated proteins and coupled this to a combined SNAP-based pulse-chase and quench-chase-pulse labeling strategy followed by high throughput imaging. We identify a host of proteins, not previously associated with CENP-A metabolism, that are involved specifically in either loading of nascent CENP-A or maintenance of the chromatin bound pool of CENP-A. We find factors with known roles in DNA replication, repair, chromatin modification and transcription. The most prominent candidate resulting from our screen is SENP6, a SUMO-specific protease, depletion of which results in strong defects in CENP-A maintenance. SENP6 is required to stabilize CENP-A chromatin throughout the cell cycle. Further, we find SENP6 to control the localization of all key centromere proteins analyzed, including CENP-C and CENP-T and the downstream kinetochore. Importantly, loss of centromere localization of CENP-A and CENP-C does not affect their cellular levels, excluding a major role for SUMO-mediated polyubiquitination and proteasomal degradation. Finally, we provide evidence for direct SUMOylation of the Constitutive Centromere-Associated Network (CCAN) components CENP-C and CENP-I that form substrates for SENP6 but not CENP-A itself.

## Results

**A screen to identify CENP-A assembly and maintenance factors.** We built upon our previous expertise to design a SNAP tag-based screening strategy to identify factors involved in CENP-A maintenance and/or assembly of new CENP-A. SNAP-pulse labeling allows for the differential labeling of either the pre-existing protein pool or of the newly synthesized pool (Fig. 1a). By using two spectrally distinct fluorophores we can track both the old pre-incorporated centromeric CENP-A pool over the course of several cell divisions and the new CENP-A pool simultaneously within the same cells and at the same centromeres (Fig. 1b).

We assembled a custom made siRNA library representing genes involved in a broad set of chromatin functions, including DNA replication, transcription, DNA repair, SUMO and Ubiquitin-regulation, nuclear organization, chromatin remodeling, and histone modifications. Our library is comprised of 2172 siRNAs encompassing 1046 genes (see Supplementary Data 1).

To screen for the involvement of these genes in CENP-A maintenance and/or assembly, we spotted 4 nanoliters of a siRNA/Lipofectamine mixture onto chambered cover glass using contact printing technology. A HeLa cell line expressing near endogenous levels of CENP-A-SNAP[10] was pulsed with a Rhodamine-conjugated SNAP substrate to label chromatin-bound CENP-A and was then quenched with a non-fluorescent SNAP substrate to block any remaining unlabeled CENP-A. These cells were solid-reverse transfected by seeding onto slides carrying 384 siRNA spots each. Following the initial pulse labeling and siRNA-mediated target mRNA depletion, cells were allowed to undergo 2 rounds of cell division and CENP-A turnover after which nascent CENP-A was pulse labeled using an Oregon green-conjugated SNAP substrate. Centromeric fluorescence intensity was determined for each siRNA condition using an automated centromere imaging pipeline (Fig. 1b, Supplementary Fig. 1, see Methods section for details).

We screened all 2172 siRNAs in 5 replicate experiments which allowed us to determine both the difference in CENP-A intensity for each siRNA relative to a scrambled control siRNA, as well as the variance in these differences. We identified 33 siRNAs (representing 31 genes) that led to a reduced CENP-A maintenance above a threshold of 1.3 fold ($-0.4$ on $\text{Log}_2$ scale) relative to controls with a significance higher than $p = 0.001$ (3 on $-\text{Log}_{10}$ scale) (Fig. 1c, Table 1, Supplementary Data 2). Among the putative genes whose depletion has a significant impact is CENP-C, which was previously found to stabilize CENP-A[9] and served as a control in our screen. In addition we find groups of genes involved particularly in (1) SUMO/Ubiquitin transactions, such as SENP6, a SUMO protease[31], (2) chromatin remodelers that include SMARCAD1, a SWI/SNF type chromatin remodeler involved in chromatin reconstitution following DNA replication[32,33], (3) chromatin modifiers such as SETD2, a H3K36 histone methyltransferase[34,35], (4) factors involved in transcription regulation, notably NACC2, a POZ domain containing protein[36] and also (5) DNA replication and repair factors such as the DNA mismatch repair protein PMS2[37]. See pathway-clustered groups of genes in Table 1. It should be noted that when proteins affect CENP-A maintenance they likely affect both ancestral, as well as newly incorporated CENP-A. We can therefore not exclude the possibility that maintenance factors play also a direct role in CENP-A assembly.

The same 2172 siRNA panel revealed a distinct set of genes when screening for factors involved in loading of nascent CENP-A (Fig. 1d and Table 2, Supplementary Data 3). We identified 42 siRNAs (representing 34 genes) that led to a reduced CENP-A assembly above a threshold of 1.9 fold ($-0.9$ on $\text{Log}_2$ scale) relative to controls with a significance higher than $p = 0.0001$ (5 on $-\text{Log}_{10}$ scale). Among these we found known assembly factors including the CENP-A specific chaperone HJURP and all members of the Mis18 complex, MIS18BP1, Mis18α and M18β, as

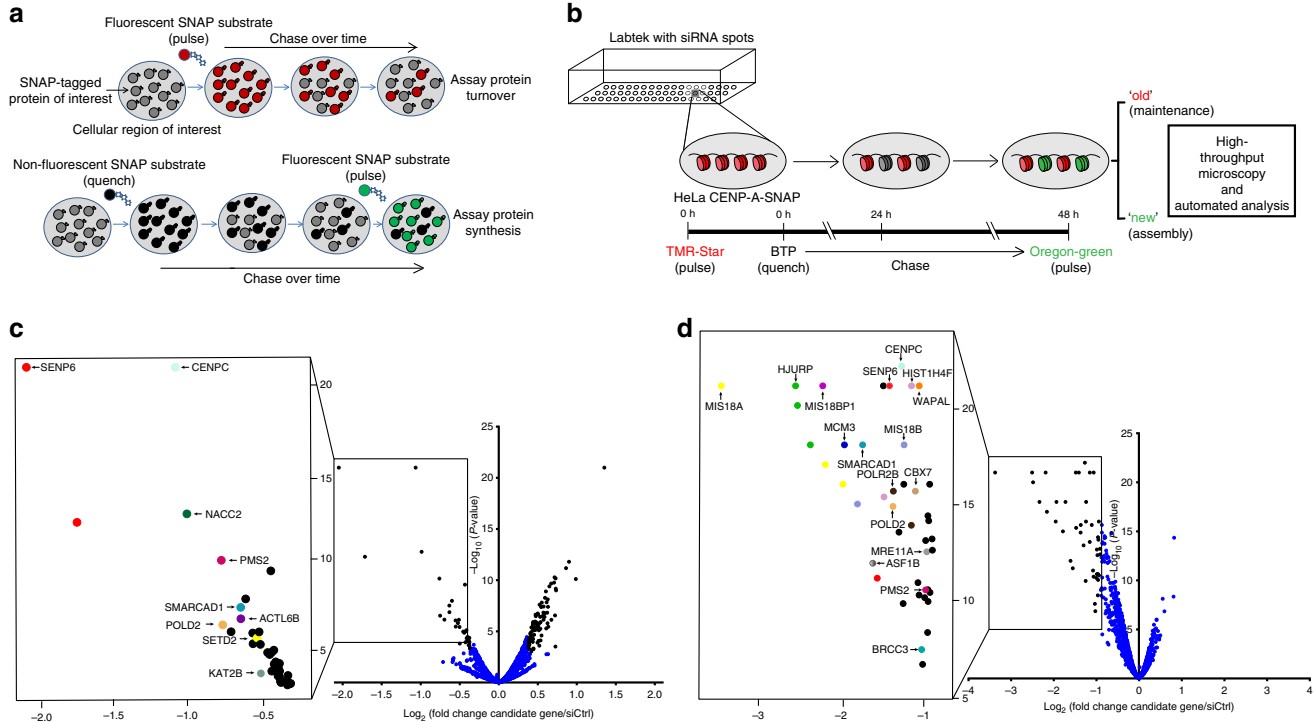

**Fig. 1 An imaging-based screen identifies CENP-A maintenance and assembly factors. a** Schematic depicting the SNAP tag-based labeling assay in order to track the maintenance and turnover (red) or de novo assembly (green) of a SNAP-tagged protein. **b** Flow scheme of high throughput siRNA screen designed to identify proteins controlling the loading or maintenance of CENP-A. HeLa CENP-A-SNAP cells, pulse labeled with TMR-Star were seeded onto chamber slides carrying 384 siRNA printed spots. During RNAi, cells were chased for 48 h followed by labeling of nascent CENP-A-SNAP with Oregon Green. CENP-A maintenance or assembly was assessed by high throughput fluorescence microscopy of red (old) or green (new) CENP-A-SNAP signals, respectively. **c** Volcano plot representing the results of the siRNA screen scoring for defects in maintenance of pre-assembled CENP-A. The fold change of mean 'old' CENP-A intensity in the candidate siRNA treatment vs. that of a negative scrambled siRNA control are plotted on the x-axis in $\log_2$ scale. The p-values of the candidate siRNA treatment as a measure of reproducibility across 5 biological replicates of the screen are plotted on the y-axis in negative $\log_{10}$ scale. The top candidate maintenance factors with a cut-off lower than a fold change ($\log_2$) of $-0.4$ and higher than a p-value ($-\log_{10}$) threshold of 3 are boxed and highlighted (identically colored dots represent different siRNA targets for the same gene). **d** Volcano plot as in **c** representing the results of the siRNA screen scoring for defects in CENP-A assembly. The top candidate assembly factors with a cut-off lower than a fold change ($\log_2$) of $-0.9$ and higher than a p-value ($-\log_{10}$) threshold of 5 are boxed and highlighted.

well as CENP-C. A recent study implicated MIS18BP1 in the maintenance of chromatin bound CENP-A[38]. However, while we find MIS18BP1 to have a major role in CENP-A assembly as previously shown[8,11,14,25], we do not detect any significant impact on CENP-A maintenance (Fig. 1c, d and Supplementary Data 2 and 3). Further we identified several components not previously associated with CENP-A assembly. Among DNA replication and repair factors we find MCM3[39] and POLD2, as well as the chromatin remodeler SMARCAD1[32] and the transcription regulator CBX7, a Polycomb repressive complex 1 (PRC1) member[40]. See pathway-clustered groups of genes in Tables 1, 2, Supplementary Data 2 and 3 and further description of the candidate factors in the discussion.

Strikingly, the top two siRNAs reducing CENP-A maintenance are both against SENP6 (4.1 and 3.3 fold reduction, 2.05 and 1.72 on $\log_2$ scale (Table 1)). SENP6 also scored high as a factor affecting CENP-A assembly (Table 2) suggesting that SENP6 is required for maintaining all CENP-A nucleosomes, including recently incorporated CENP-A. Alternatively, it has an additional role specifically in assembly.

**SENP6 controls CENP-A maintenance.** The SUMO protease SENP6 is an enzyme involved in proteolytic removal of SUMO2/3 chains from target proteins[31]. SENP6 is known to be involved in various processes including inflammatory signaling and DNA repair[41,42]. Furthermore, SENP6 has previously been implicated

in regulating the levels of CENP-H and CENP-I at the centromere[43]. To explore the involvement of SENP6 in CENP-A regulation we depleted SENP6 from HeLa CENP-A-SNAP cells by siRNA. We confirmed at high resolution and single centromere intensity measurements that SENP6 affects maintenance of both the CENP-A pool that was incorporated into chromatin prior to SENP6 depletion (Fig. 2a, c, e), as we all as maintenance of the nascent CENP-A pool (Fig. 2b, d, e). To address the possibility that the effect of SENP6 on CENP-A maintenance is affected by the SNAP tag or the expression levels of the transgene, we performed siRNA mediated depletion of SENP6 in unmodified HeLa cells, as well as in primary human fibroblasts and measured the centromeric levels of endogenous CENP-A. The results confirmed that SENP6 is a bona fide regulator of centromeric CENP-A levels (Fig. 2f-i) both in cancer and primary cells.

**SENP6 is continuously required to maintain CENP-A chromatin.** A critical barrier for chromatin maintenance is S phase during which DNA replication disrupts histone DNA contacts. A key step in the stable propagation of chromatin is histone recycling during DNA replication involving the MCM2-7 helicase complex, along with dedicated histone chaperones[44], a role recently extended to the stable transmission of CENP-A nucleosomes at the replication fork[45]. Further, recent work has shown that the CENP-A chaperone HJURP is required to recycle

**Table 1 List of candidate genes that affect the maintenance of 'old' CENP-A at the centromere.**

| Gene name | siRNA id | Fold difference vs. control* | Significance $p$ value | Biological function | Pathway |
|---|---|---|---|---|---|
| CENPC | s2913 | −1.06504 | 0 | Kinetochore assembly | Centromere and kinetochore assembly |
| CENPW | s51837 | −0.4414 | 1.36E−05 | Kinetochore assembly | |
| AURKB | s17612 | −0.43346 | 2.74E−10 | Aurora kinase B, mitotic error correction | |
| CENPI | s5374 | −0.42588 | 1.01E−05 | Kinetochore assembly | |
| SENP6 | s25025 | −2.04927 | 0 | SUMO-specific endopeptidase | SUMO/Ubiquitin |
| SENP6 | s25024 | −1.71502 | 5.14E−13 | SUMO-specific endopeptidase | |
| UBE2A | s14565 | −0.55306 | 8.57E−07 | Ubiquitin-conjugating enzyme E2 A | |
| KEAP1 | s18981 | −0.55133 | 3.42E−06 | Cul3-RING ubiquitin ligase complex | |
| SMARCAD1 | s32385 | −0.63277 | 3.06E−08 | ATP dependent chromatin remodeler | Chromatin remodelers |
| ACTL6B | s28105 | −0.63177 | 1.35E−07 | Chromatin remodeler | |
| CHD8 | s33580 | −0.58555 | 0.006905 | Chromodomain-helicase | |
| CHD8 | s33581 | −0.4108 | 0.007693 | Chromodomain-helicase | |
| HLTF | s13138 | −0.40628 | 0.006822 | SWI/SNF Related helicase-like TF | |
| SUV420H2 | s195487 | −0.71378 | 0.009868 | Histone H4-K20 trimethylation | Chromatin modifiers |
| EZH2 | s4918 | −0.59289 | 0.009957 | Histone H3 K27 methyl transferase | |
| HDAC4 | s18838 | −0.57779 | 0.004177 | Histone deacetylase | |
| SETD2 | s26423 | −0.52355 | 1.58E−06 | Histone H3 K36 methyl transferase | |
| KAT2B | s16895 | −0.49978 | 0.000157 | Histone acetyltransferase PCAF | |
| SMYD1 | s45456 | −0.45007 | 0.001412 | Histone-lysine N-methyltransferase | |
| EHMT2 | s21469 | −0.40194 | 7.69E−05 | Histone H3 K27 methyl transferase | |
| NCOA1 | s16461 | −0.4005 | 3.54E−05 | Nuclear receptor coactivator 1 (HAT) | Transcription regulation |
| NACC2 | s44088 | −0.9889 | 1.70E−13 | Transcription corepressor | |
| POLR2B | s10796 | −0.5025 | 3.74E−06 | RNA polymerase II subunit | |
| CDK9 | s2835 | −0.45452 | 1.06E−05 | Cyclin-dependent kinase 9 | |
| ARID4B | s28603 | −0.42745 | 0.000117 | Transcriptional corepressor | |
| MCM3 | s8591 | −0.51112 | 7.42E−07 | DNA replication initiation | DNA replication and repair factors |
| PMS2 | s10741 | −0.76103 | 6.93E−11 | Mismatch repair specific endonuclease | |
| POLD2 | s10779 | −0.75101 | 2.95E−07 | DNA polymerase delta | |
| NUP54 | s28724 | −0.69635 | 7.17E−07 | Nuclear pore complex protein | Miscellaneous |
| HIST1H2AA | s48083 | −0.60008 | 1.04E−08 | Histone H2A type 1-A | |
| SYNE1 | s23608 | −0.58868 | 0.005326 | Nuclear envelope protein | |
| SMC1A | s15751 | −0.41301 | 7.84E−06 | Cohesion complex | |
| PADI2 | s22189 | −0.41185 | 4.70E−05 | Protein-arginine deiminase type-2 | |

*$\log_2$ (fold change of candidate gene vs. control)
List of candidate genes affecting maintenance of pre-assembled centromeric CENP-A, clustered in pathways. Listed are hits with a fold difference higher than 1.3 (−0.4 on $\log_2$ scale) of mean 'old' CENP-A signal intensity in the siRNA treatment vs. the negative scrambled siRNA control and have a significance over <0.001

CENP-A specifically during S phase[45]. In our screen, HJURP is not a prominent contributor to overall CENP-A maintenance, relative to SENP6 or other candidate maintenance factors (Supplementary Data 2). Nevertheless, S phase may be a critical cell cycle window where CENP-A maintenance requires the re-assembly of pre-existing CENP-A chromatin onto nascent DNA. To determine whether SENP6 plays a role in CENP-A recycling and maintenance of centromere integrity during DNA replication or has broader roles along the cell cycle we constructed a SENP6 specific auxin-inducible degron (AID)[46] that has been successfully exploited previously to address the acute role of centromere proteins[29,45,47–49].

We targeted all SENP6 alleles in HeLa cells, which express the CENP-A-SNAP transgene, as well as the Oryza sativa-derived E3 ligase, TIR1 with a miniAID-GFP construct (Fig. 3a). Addition of the auxin Indole-3-acetic acid (IAA) resulted in rapid loss of SENP6 eliminating the majority of the nuclear pool within 3 h (Supplementary Fig. 2A, B). Longer exposure to IAA resulted in cell growth arrest confirming SENP6 to be an essential protein for cell viability (Supplementary Fig. 2C). In agreement with the siRNA experiments above, SENP6 degradation over a 24-48 h period led to a loss of CENP-A from centromeres in SNAP-based pulse-chase measurements (Fig. 3b, c, Supplementary Fig. 2D). Strikingly, time course experiments of IAA addition showed that loss of CENP-A becomes evident within 6 h of SENP6 depletion (Fig. 3d). The acute effect of SENP6 depletion on CENP-A

nucleosomes enables us to determine at what stage during the cell cycle CENP-A stability depends on SENP6 action.

To this end, we synchronized HeLa CENP-A-SNAP expressing cells at the G1/S boundary using thymidine, pulse labeled CENP-A-SNAP followed by release into S phase (Fig. 3e). We depleted SENP6 by the addition of IAA for 6 h specifically during S phase or after entry into G2 phase during an arrest with the Cdk1 inhibitor for 4 h. The degree of synchronization was assessed by measuring the percentages of EdU positive (80%) and cyclin B positive (87.9%) cells representing S phase and G2 phase synchronized populations, respectively. To assess the contribution of SENP6 to CENP-A maintenance in G1 phase we synchronized cells in G2 phase with the Cdk1 inhibitor RO3306 and depleted SENP6 following the release from the inhibitor in mitosis and G1. Irrespective of whether SENP6 was depleted in S, G2, or G1 phase, CENP-A was lost from the centromere (Fig. 3f–h). This is a striking observation and indicates that CENP-A chromatin is under continued surveillance and at risk of loss, presumably by SUMO modification. It highlights that CENP-A turnover is a process not only coupled to DNA replication but to other cellular processes as well that may include transcription or other chromatin remodeling activities. Similarly, when cells were synchronized in G1 phase by mitotic arrest and release (Fig. 3i) and newly synthesized CENP-A was labeled we found that freshly assembled CENP-A cannot be maintained in the absence of SENP6 (Fig. 3j). While SENP6 is

**Table 2 List of candidate genes that affect the loading of 'new' CENP-A at the centromere.**

| Gene name | siRNA id | Fold difference vs. control* | Significance p value | Biological function | Pathway |
|---|---|---|---|---|---|
| *MIS18A* | s28851 | −3.37055 | 0 | CENP-A assembly | CENP-A nucleosome assembly |
| *MIS18A* | s28852 | −2.1571 | 0 | CENP-A assembly | |
| *MIS18A* | s28853 | −1.94987 | 0 | CENP-A assembly | |
| *HJURP* | s30815 | −2.50416 | 0 | CENP-A assembly | |
| *HJURP* | s30814 | −2.48024 | 0 | CENP-A assembly | |
| *HJURP* | s30813 | −2.33028 | 0 | CENP-A assembly | |
| *MIS18BP1* | s30722 | −2.18629 | 0 | CENP-A assembly | |
| *MIS18BP1* | s30720 | −0.90609 | 2.23E−13 | CENP-A assembly | |
| *MIS18β*$^{(OIP5)}$ | s22368 | −1.78149 | 0 | CENP-A assembly | |
| *MIS18B*$^{(OIP5)}$ | s225496 | −1.23665 | 0 | CENP-A assembly | |
| *CENPC* | s2913 | −1.26733 | 0 | kinetochore assembly | Centromere and kinetochore assembly |
| *CENP-R*$^{(ITGB3BP)}$ | s23796 | −1.47942 | 0 | kinetochore assembly | |
| *SENP6* | s25024 | −1.54918 | 5.83E−12 | SUMO-specific endopeptidase | SUMO/Ubiquitin |
| *SENP6* | s25025 | −1.40764 | 0 | SUMO-specific endopeptidase | |
| *BRCC3* | s35699 | −1.03194 | 2.44E−08 | Lys63-specific deubiquitinase, positive regulation of DNA repair | |
| *SMARCAD1* | s32385 | −1.72173 | 0 | ATP-dependent chromatin remodeling | Chromatin remodelers |
| *SMARCD3* | s13158 | −0.98509 | 7.13E−14 | ATP dependent chromatin remodeler | |
| *ASF1B* | s31344 | −1.60186 | 1.02E−12 | Replication-dependent nucleosome assembly | |
| *SMYD2* | s32468 | −0.96248 | 3.37E−09 | Histone-lysine N-methyltransferase | Chromatin modifiers |
| *NSD2*$^{(WHSC1)}$ | s200461 | −0.9385 | 0 | Histone H3K27 methyl transferase activity | |
| *SUV39H2* | s36183 | −0.90985 | 5.97E−14 | Histone H3K9 methyl transferase activity | |
| *POLR2B* | s10798 | −1.36259 | 2.22E−16 | RNA polymerase II subunit RPB2 | Transcription regulation |
| *POLR2B* | s10796 | −1.15233 | 1.20E−14 | RNA polymerase II subunit RPB2 | |
| *GTF2H4* | s6318 | −1.29604 | 2.71E−14 | General transcription factor IIH subunit 4 | |
| *CBX7* | s23926 | −1.10529 | 2.22E−16 | Polycomb group complex, transcription repression | |
| *L3MBTL* | s24934 | −0.95948 | 2.27E−11 | Polycomb group protein, transcription repression | |
| *MYC* | s9130 | −0.95904 | 4.00E−15 | Activating transcription factor | |
| *BRD2* | s12071 | −0.9572 | 8.92E−11 | Regulation of transcription by RNA polymerase II | |
| *MCM3* | s8591 | −1.93285 | 0 | DNA replication initiation | DNA replication and repair factors |
| *POLD2* | s10779 | −1.36601 | 1.33E−15 | DNA polymerase delta subunit 2 | |
| *PMS2* | s10741 | −0.98535 | 2.29E−11 | Mismatch repair specific endonuclease | |
| *MRE11A* | s8960 | −0.97116 | 2.69E−13 | 3'–5' exonuclease activity, DNA double-strand break processing | |
| *HIST1H4L* | s15897 | −1.47046 | 4.44E−16 | Histone H4 | Miscellaneous |
| *HIST1H2BF* | s15859 | −1.24834 | 1.13E−10 | Histone H2B | |
| *SYNE1* | s23608 | −1.24079 | 0 | Nuclear envelope protein | |
| *HIST1H4F* | s15886 | −1.14982 | 0 | Histone H4 | |
| *HCFC1* | s6476 | −1.07546 | 9.84E−12 | Cell cycle regulation, host-virus interaction | |
| *TOX4* | s19129 | −1.06299 | 4.21E−11 | PTW/PP1 phosphatase complex | |
| *WAPAL* | s22949 | −1.05942 | 0 | Negative regulator of sister chromatid cohesion | |
| *MIOS* | s29027 | −0.99597 | 5.63E−11 | Positive regulation of TOR signaling | |
| *KIF2B* | s39236 | −0.94892 | 7.11E−15 | Microtubule motor | |
| *PPP2R1A* | s10964 | −0.93523 | 3.09E−11 | Protein phosphatase 2 A subunit, chromosome segregation | |

*log$_2$ (fold change of candidate gene vs. control)
List of candidate genes affecting assembly of nascent CENP-A at centromeres, clustered in pathways. Listed are hits with a fold difference higher than 1.9 (−0.9 on log$_2$ scale) of mean 'new' CENP-A signal intensity in the siRNA treatment vs. the negative scrambled siRNA control and have a significance over <0.0001

likely responsible for the maintenance of newly loaded CENP-A, our experiments cannot rule out a direct role for SENP6 in CENP-A assembly.

We confirmed these results in unsynchronized cells in which we depleted SENP6 for 7 h and analyzed CENP-A loss at specific stages of the cell cycle (Supplementary Fig. 3). We scored G1 phase cells by the presence of α-tubulin positive midbodies, S phase cells by EdU pulse labeling to mark for active DNA replication, and G2 phase cells by cytosolic cyclin B staining. SENP6 depletion resulted in loss of CENP-A irrespective of cell cycle position. These results indicate that SENP6 is continuously

required throughout the cell cycle to prevent CENP-A from being removed from the centromere.

**CENP-A maintenance requires catalytic activity of SENP6.** We next determined whether the SENP6 isopeptidase activity is required to ensure CENP-A maintenance at the centromere. We transfected a construct expressing wild-type or a catalytic mutant SENP6$^{C1030A}$ into HeLa cells expressing CENP-A-SNAP and coupled it to auxin mediated depletion of the endogenous SENP6 protein. Under these conditions we followed the fate of

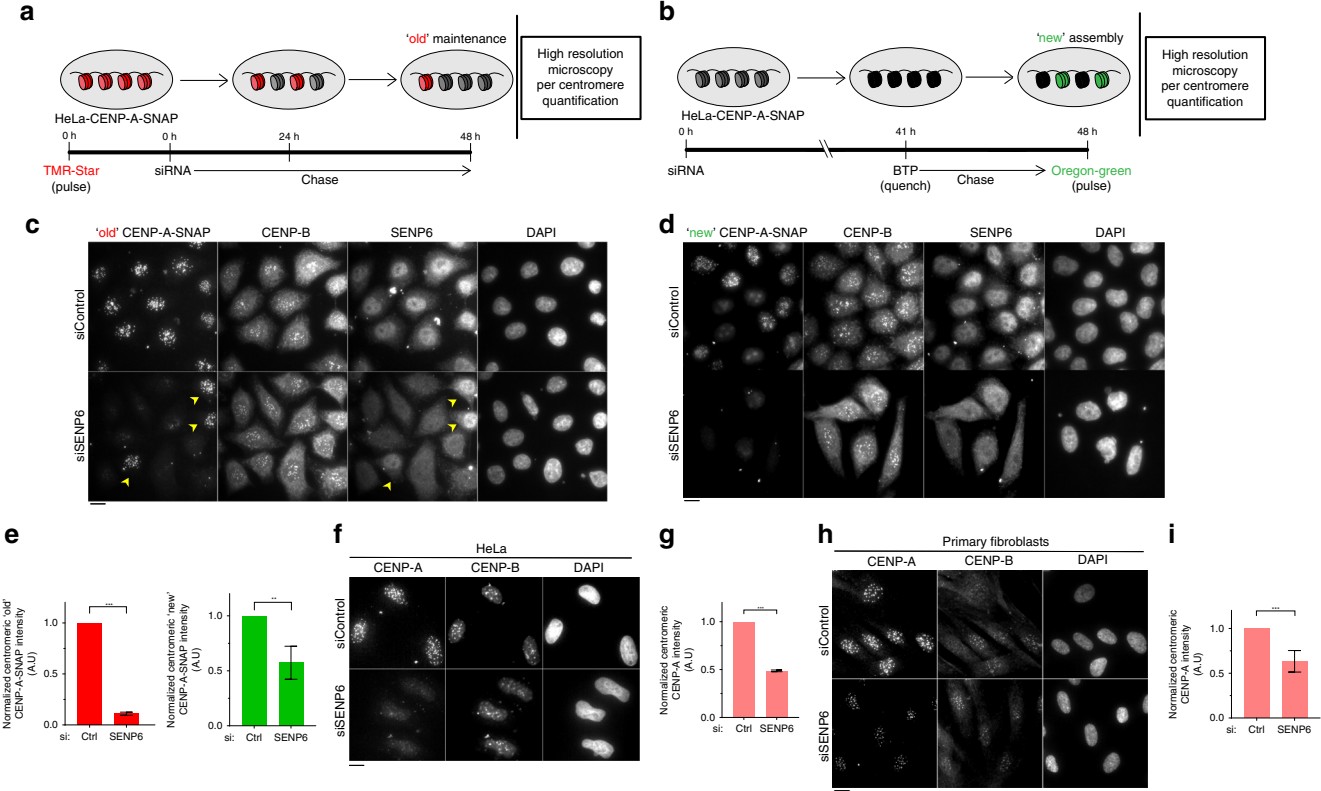

**Fig. 2 SENP6, a SUMO protease, controls maintenance of centromeric CENP-A. a**, **b** Schematics for high resolution SNAP pulse-chase (**a**) and quench-chase-pulse (**b**) assays. HeLa-CENP-A-SNAP cells were treated with *SENP6* siRNA or a control scrambled siRNA. Pulse-chase experiment was performed for 48 h during RNAi to assay for CENP-A turnover (**a**). Quench-chase-pulse experiment was performed for the final 7 h of siRNA treatment to assay for CENP-A assembly (**b**). **c**, **d** shows typical image fields following the strategies in **a**, **b**, respectively. TMR-Star and Oregon Green SNAP labels visualize the maintenance or assembly of CENP-A-SNAP, respectively. CENP-B was used as a centromeric reference for quantification. Cells were counterstained for SENP6 to visualize its depletion in siRNA treated cells. Yellow arrowheads indicate nuclei that escaped SENP6 depletion which correlate with retention of 'old' CENP-A-SNAP. Bars, 10 μm. **e** Automated centromere recognition and quantification of **c**, **d**. Centromeric CENP-A-SNAP signal intensities were normalized to the control siRNA treated condition in each experiment. siRNA treatment; siSENP6 or scrambled (Ctrl). Three replicate experiments were performed. Bars indicate SEM. Parametric two-tailed Student's *t* test were performed to calculate statistical significance. **p < 0.01, ***p < 0.001. **f** Centromeric levels of total, steady state CENP-A under siRNA mediated depletion of SENP6 in untagged HeLa cells. **g** Quantification of **f** as in **c**, **d**. **h** Centromeric levels of total, steady state CENP-A under siRNA mediated depletion of SENP6 in primary fibroblasts. **i** Quantification of **h** as in **c** and **d**. Source data are provided as a Source Data file.

pulse-labeled centromeric CENP-A-SNAP (Fig. 4a–c). While wild-type SENP6 overexpression rescues the defect in centromeric CENP-A maintenance, the catalytic dead mutant was unable to do so, indicating that centromeric retention of CENP-A is directly regulated by the desumoylation activity of SENP6.

**CENP-A depleted centromeres are competent for CENP-A re-assembly**. CENP-A assembly depends on a functional CCAN[6,11,14,50] and is ultimately dependent on pre-existing centromeric nucleosomes. This raises the question whether the severe loss of CENP-A from the centromere impedes assembly of new CENP-A. To answer this, we depleted AID-SENP6 by auxin for 24 h, inducing CENP-A loss, followed by a quench-chase-pulse assay to score for de novo assembly of CENP-A (Fig. 4e, f). During the time of new CENP-A synthesis we washed out the auxin to allow for SENP6 re-expression for 48 h (Fig. 4d). We also maintained 'no-auxin' and 'continued auxin' conditions as positive and negative controls for CENP-A assembly, respectively. We found that SENP6 depletion severely compromised the loading of new CENP-A as the percentage of TMR positive cells dropped to less than 5% of the population as compared to 80% cells in the 'no auxin' condition. However, upon SENP6 re-expression, the percentage of TMR positive cells increased to 40%, indicating

nascent CENP-A can be targeted to compromised centromeres, at least partially, once SENP6 action is reinstalled (Fig. 4e, f).

**SENP6 stabilizes multiple centromere and kinetochore proteins**. Next, we determined whether other centromere components were also affected by depletion of SENP6. Previous reports have indicated a requirement of SENP6 for maintaining CENP-H and CENP-I at centromeres[43]. We now found that in addition to CENP-H and CENP-I, the major structural centromere components CENP-C, CENP-T, along with CENP-A are strongly reduced after auxin-mediated depletion of SENP6 (Fig. 5a, b, d). These findings indicate that the entire DNA-associated foundation of the centromere is under the critical control of SENP6. A notable exception is CENP-B, an alpha-satellite binding centromere protein, which remains unaffected by SENP6 depletion (Fig. 5b, d). Furthermore, members of the microtubule binding KMN-network, NNF1, DSN1, and HEC1 are also reduced after SENP6 depletion (Fig. 5c, d). We corroborated this data using siRNA mediated depletion of SENP6 in untagged HeLa cells that resulted in loss of CENP-C, CENP-I, and CENP-T (Supplementary Fig. 4A, B).

It has been shown that polySUMOylation can lead to subsequent polyubiquitination by SUMO-directed ubiquitin

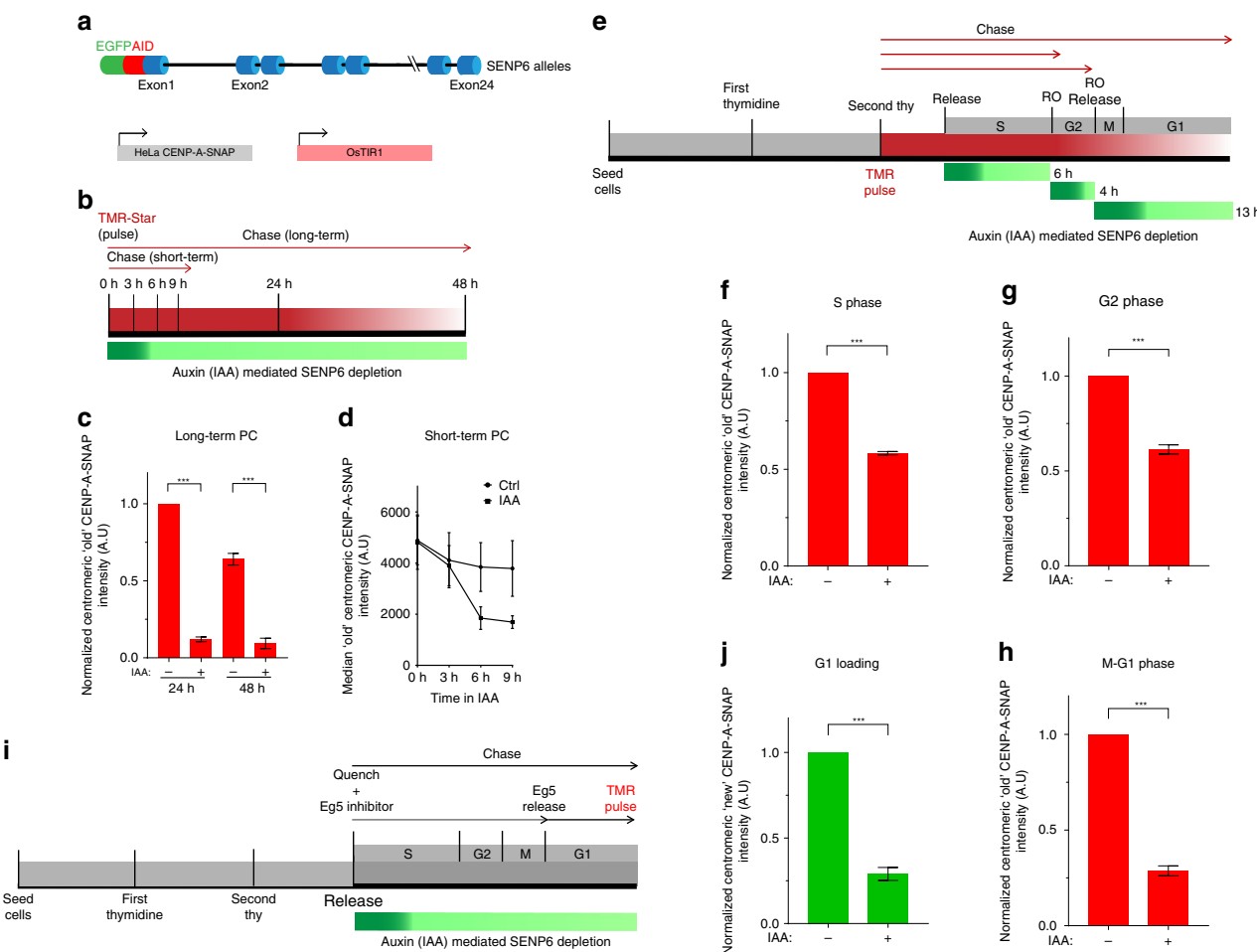

**Fig. 3 SENP6 is required for centromeric CENP-A maintenance throughout the cell cycle. a** Schematic of the genotype of cell line constructed for auxin (IAA)-mediated depletion of SENP6. OsTIR1 and CENP-A-SNAP are expressed as transgenes, *SENP6* is homozygously tagged at its endogenous locus. **b** Experimental scheme for long-term and short-term CENP-A-SNAP pulse-chase (PC) assays following auxin (IAA) mediated depletion of SENP6. **c, d** Quantification of long-term and short-term PC experiments, respectively following the experimental scheme detailed in **b**. 'Old' centromeric CENP-A-SNAP intensities are normalized to the mean of the non-treated condition (−) for the 24 h time point and plotted as bar graphs against auxin (IAA) treated (+) and non-treated (−) conditions for 24 h and 48 h. Three replicate experiments were performed. Bar indicates SEM. Parametric two-tailed Student's t test were performed to calculate statistical significance. ***$p < 0.001$ **d** 'Old' centromeric CENP-A-SNAP intensities are measured and median intensities are plotted against auxin (IAA) or non-treated (Ctrl) conditions. Bars indicate SEM of three replicate experiments. **e** Experimental scheme of cell cycle synchronization coupled to CENP-A-SNAP pulse-chase (TMR pulse) and auxin (IAA) mediated depletion of SENP6 at different stages of the cell cycle (RO: RO3306, Cdk1 inhibitor). **f–h** Quantification as in **c** but normalized to non-treated condition (−) of 'old' CENP-A-SNAP intensities after auxin (IAA) mediated depletion of SENP6 at different stages of the cell cycle following the experimental scheme of **e**. **i** Experimental scheme of cell cycle synchronization in G1 phase coupled to CENP-A-SNAP quench-chase-pulse and auxin (IAA) mediated depletion of SENP6. EG5 inhibitor Dimethylenastron was used for a mitotic arrest followed by release into G1. **j** Quantification as in **f–h** of 'new' CENP-A-SNAP intensities in G1 stage of the cell cycle after auxin (IAA) mediated depletion of SENP6 following the experimental scheme in **i**. Source data are provided as a Source Data file.

ligases such as RNF4. This mechanism was previously implicated in the degradation of CENP-I[43]. As a deSUMOylase, SENP6 may prevent degradation of CENP-I and other centromere/kinetochore proteins by removing SUMO chains. Indeed, SUMO-mediated polyubiquitination and subsequent degradation of budding yeast CENP-A[Cse4] has been reported[51,52]. Therefore, we measured the protein stability of CENP-A and CENP-C after depletion of SENP6. Unexpectedly, we find that while these proteins are depleted from centromeres, cellular levels of both proteins are maintained upon SENP6 depletion, either by auxin-mediated degradation (Fig. 5e) or siRNA (Supplementary Fig. 4C). This suggests that hyper-SUMOylation of target proteins in SENP6 depleted cells does not lead to polyubiquitination and proteasomal degradation but rather that a SUMO cycle controls localization.

As chromatin-bound CENP-A which is removed from the centromere in SENP6 depleted cells is not degraded, it is potentially available for re-assembly. To test this, we pulse labeled centromeric CENP-A, and followed the fate of this pool during auxin mediated depletion of AID-SENP6 for 24 h followed by an auxin washout, allowing for SENP6 re-expression for 48 h (Fig. 4g). We found that, in contrast to nascent CENP-A, the pool that is displaced from centromeric chromatin is not recycled back to the centromere (Fig. 4h). This indicates a functional difference between the nascent and ancestral pools in their ability to be recognized by the assembly machinery.

**SENP6 controls polySUMOylation of the centromere.** SENP6 depletion leads to the polySUMOylation of its substrates in a

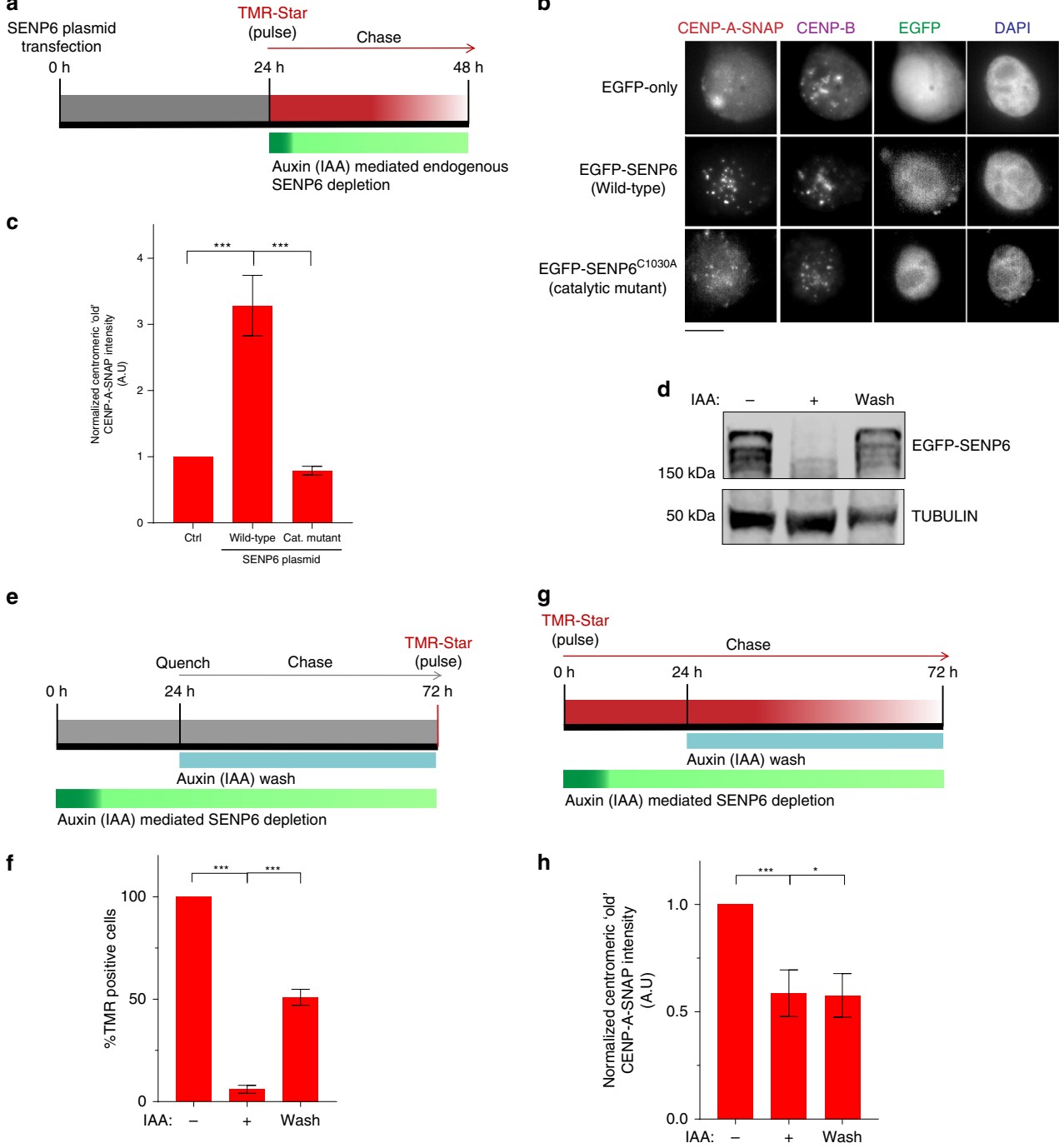

**Fig. 4 CENP-A maintenance requires SENP6 catalytic activity. a** Experimental scheme coupling overexpression of wild-type or catalytic mutant SENP6 to auxin mediated depletion of SENP6 and tracking the 'old' CENP-A-SNAP by pulse-chase (PC) assay. **b** Images of remaining pre-incorporated CENP-A-SNAP signal following auxin mediated depletion of endogenous SENP6 and over-expression of EGFP only (Ctrl), wild-type SENP6 or catalytically dead SENP6[C1030A]. Bars, 10 μm. **c** Quantification of the experiment as outlined in **a**. 'Old' centromeric CENP-A-SNAP intensities were normalized to the mean of the control condition (Ctrl). Three replicate experiments were performed. Bars represent SEM. Parametric two-tailed Student's *t* test was performed to calculate statistical significance. ***$p < 0.001$. **d** Immunoblot showing the levels of AID-SENP6 by auxin treatment (24 h) and re-expression following auxin wash for 48 h. **e, g** Experimental schemes coupling auxin mediated depletion and re-expression of SENP6 by auxin wash to pulse-chase and quench-chase-pulse assay for tracking pre-maintained CENP-A or newly loaded CENP-A. **f, h** Quantification of the above. Three replicate experiments were performed. Bars represent SEM. Parametric two-tailed Student's *t* test was performed to calculate statistical significance. *$p < 0.05$, ***$p < 0.001$. Source data are provided as a Source Data file.

SUMO2/3 dependent manner[53]. We hypothesized that CCAN components represent SENP6 substrates which, upon SENP6 loss would be expected to be hyper-SUMOylated, leading to their delocalization. In order to test our hypothesis, we pulled down

His10-tagged SUMO2 under auxin or control condition in AID-SENP6 expressing HeLa cells and immunoblotted for representative CCAN and CENP-A assembly components (Fig. 6a–f). We found poly-SUMOylated species of both CENP-C and

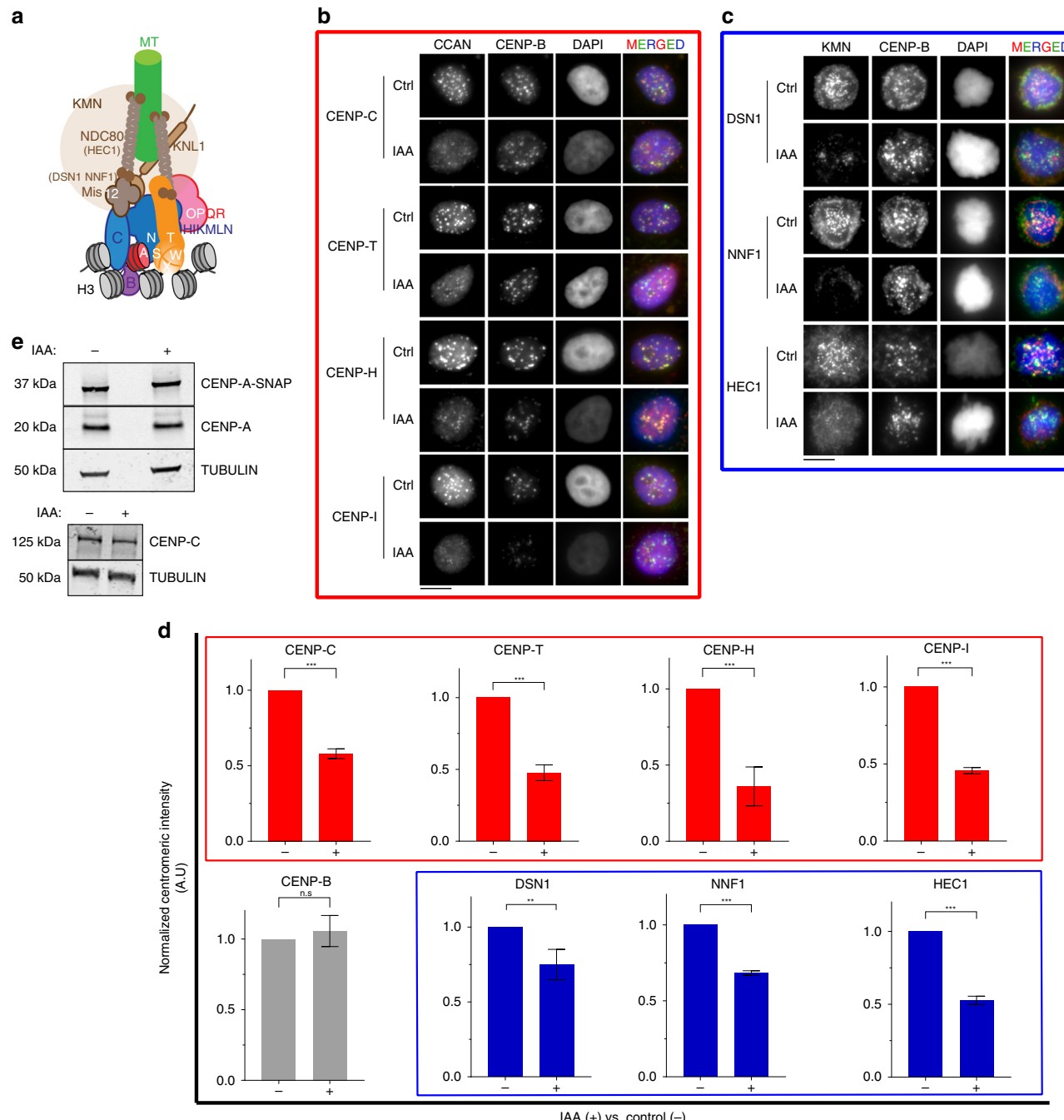

**Fig. 5 SENP6 is required for maintaining the integrity of the CCAN independent of proteolysis. a** Schematic representing the architecture and interactions of different protein complexes in the human centromere and kinetochore. **b** Centromeric levels of different CCAN proteins following auxin-mediated depletion of SENP6 (24 h). Cells were counterstained with CENP-B to mark centromeres. Bars, 10 μm. **c** Centromeric levels of KMN (KNL1-MIS12-NDC80) members following auxin-mediated depletion of SENP6. Bars, 10 μm. **d** Automated centromere recognition and quantification of **b** and **c**. Fluorescence intensities of indicated proteins were normalized to the mean of the control 'no auxin' condition in each experiment and plotted as a bar graph for auxin treated '+' or no auxin '−' condition. Three replicate experiments were performed. Bars represent SEM. Parametric two-tailed Student's $t$ test were performed to calculate statistical significance. **$p < 0.01$, ***$p < 0.001$. **e** Immunoblot showing the total levels of CENP-A and CENP-C proteins following 48 h treatment with auxin to deplete AID-SENP6. Extracts from control '−' or auxin treated '+' cells were separated by SDS-PAGE and immunoblotted with anti-CENP-A (detecting both SNAP tagged and endogenous CENP-A) or anti-CENP-C antibody. Tubulin was used as loading control. Source data are provided as a Source Data file.

CENP-I, that accumulated in manner dependent on SENP6 loss, suggesting these proteins are bona fide SENP6 targets. In contrast, we could detect only faint or no poly-SUMOylation for both CENP-A and MIS18BP1. Two studies coincident to ours also reported poly-SUMOylation of multiple CCAN subunits upon siRNA mediated depletion of SENP6, discovered through mass spectrometry analysis[54,55]. Our results, along with these papers provide evidence for CCAN SUMOylation but no direct modification of CENP-A, indicating it is not a direct substrate for SENP6. Rather, the maintenance of CENP-A may result from

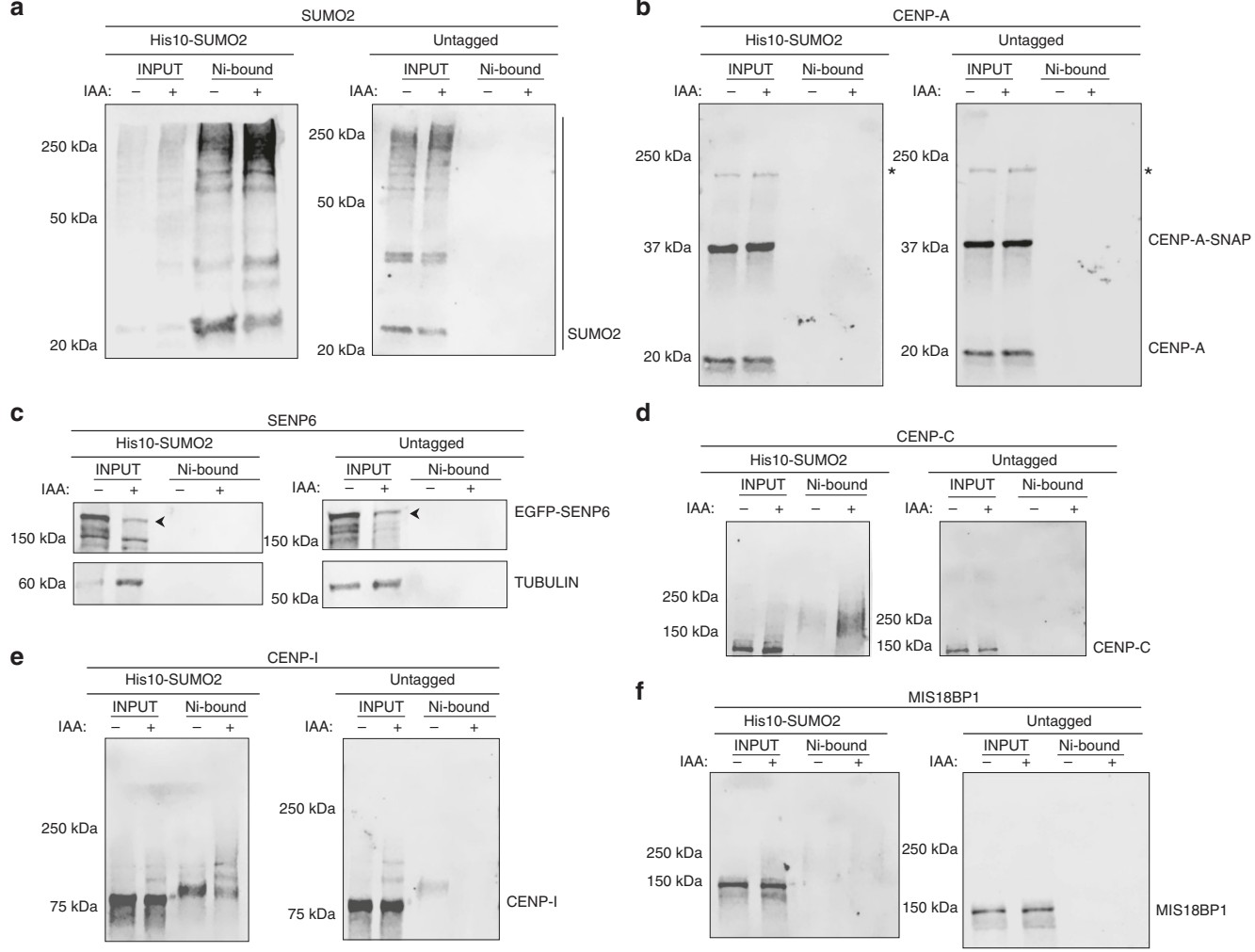

**Fig. 6 CCAN subunits are polySUMOylated in absence of endogenous SENP6.** SUMOylation targets of SENP6 in the inner kinetochore revealed by Immuno blotting of SDS-PAGE-separated His10-SUMO2 pull-downs from extracts of cells that were subjected to either control or auxin-mediated depletion of SENP6. **a** Blot probed for SUMO2 in control '−' vs. auxin treated '+' condition in His10-SUMO2 overexpressing cells and the parental untagged cells. **b** Blot as in **a** probed for CENP-A. Asterisk (*) indicates non-specific band. **c** Blot as in **a** probed for SENP6. Arrowhead indicates the EGFP-SENP6 position. **d** Blot as in **a** probed for CENP-C. Note the His10-SUMO2 and SENP6 depletion-dependent accumulation of CENP-C-positive high molecular weight species. **e** Blot as in **a** probed for CENP-I. Note the His10-SUMO2 and SENP6 depletion-dependent accumulation of CENP-I-positive high molecular weight species. **f** Blot as in **a** probed for MIS18BP1. Source data are provided as a Source Data file.

deSUMOylation, and the resulting stabilization of CCAN subunits such as CENP-C, which are known to be critical for maintaining the stability of centromeric CENP-A.

## Discussion

The CENP-A nucleosome is a key determinant of the heritable maintenance of centromeres. A key question that remains is how is CENP-A assembled into chromatin and perhaps more enigmatically, how is it stably transmitted across multiple cell division cycles, a property central to the epigenetic propagation of the centromere. To what extent is stable transmission an intrinsic property encoded within the CENP-A histone or is stable chromatin binding imposed by external factors? To gain insight, we devised a comprehensive screen that is specifically designed to identify nuclear chromatin-associated proteins required for maintaining the chromatin-bound pool of CENP-A. We identified a series of putative factors not previously associated with CENP-A dynamics and centromeric chromatin maintenance.

Among the top ranking candidates we find several clusters of functionalities involved in CENP-A chromatin maintenance. Factors that stand out are proteins involved in centromere and kinetochore function that includes CENP-C as expected[9,28] but also CENP-W and CENP-I, as well as the mitotic kinase Aurora B, indicating that several layers within the centromere and kinetochore impact on centromeric chromatin maintenance.

In addition, chromatin remodeling factors or members of ATP dependent motors are among the list which includes SMARCAD1, part of the SNF subfamily of helicase proteins, which has been implicated in nucleosome turnover at sites of repair[56]. We find both previously incorporated, as well as new CENP-A to be affected by loss of SMARCAD1. The related protein SMARCD3[BAF60C][57] was identified as well to specifically affect the nascent pool of CENP-A, possibly implicating it in CENP-A assembly. Interestingly, the fission yeast homolog of SMARCAD1, fft3 has been shown to be involved in faithful maintenance of heterochromatin[33] by reducing histone turnover. Possibly it plays an analogous function in CENP-A chromatin maintenance. Other chromatin remodelers include

ACTL6B[BAF53B], an actin-related protein that is a subunit of the BAF (BRG1/brm-associated factor) complex[58] in mammals, which is functionally related to SWI/SNF chromatin remodeler complexes[59]. HLTF[SMARCA3], another SWI/SNF family member[60] and CHD8 (Chromodomain Helicase DNA Binding Protein 8)[61] also have a significant impact on CENP-A maintenance.

Related to the chromatin remodelers are factors involved in transcriptional repression, NACC2 (also known as RBB) which contains a POZ domain and recruits the NuRD (Nucleosome Remodeling Deacetylase) complex for gene silencing[36], and ARID4B[62] a subunit of the histone deacetylase-dependent SIN3A transcriptional corepressor complex. General components of the transcription machinery such as the POLR2B, a subunit of RNA polymerase II and CDK9[P-TEFb], a critical kinase in the transcription elongation complex[63] also impact on CENP-A maintenance, as well as assembly (for POLR2B). Pleiotropic effects of depleting these general transcription-related proteins cannot be excluded but nevertheless these factors are of interest as a direct role for transcription in CENP-A maintenance and assembly has been suggested[64–66].

Furthermore, in addition to core transcription components and chromatin remodelers we identified a series of chromatin modifying enzymes including SUV420H2, a histone H4K20 methyltransferase[67], EZH2, the PRC2 complex component responsible for H3K27 methylation[68], SETD2 the principal H3K36 methyl transferase[34,35], EHMT2 (G9A), a H3K9 specific methyltransferase[69] and SMYD1 an SET domain protein that potentially targets histones[70]. The variety of modifiers that impact on CENP-A maintenance suggests that a chromatin imbalance, whether it is activating or repressing for gene expression, has a deleterious effect on CENP-A chromatin maintenance. SUV420H2 is of particular interest as histone H4 in the context of CENP-A nucleosomes has been shown to be monomethylated at lysine 20 and impact on centromere structure[71]. It would be of interest to determine whether SUV420H2 is responsible for this modification and can affect CENP-A nucleosome stability directly. In addition, we identified two acetyl transferases with a potential role in maintaining CENP-A levels. These are NCOA1 and KAT2B [also known as PCAF, a component of the p300 complex[72]], as well as an deacetylase, HDAC4.

Our screening setup allows us to differentiate between proteins involved in recruiting CENP-A to the centromere and those that are required to maintain CENP-A in chromatin, once incorporated. However some components that we identified play a role in both. Possibly, this reflects a requirement of those factors in CENP-A maintenance, both CENP-A that was previously incorporated, as well as newly incorporated CENP-A. Factors in this category include CENP-C as has been reported before[9] but also the DNA replication factors POLD2 (a DNA polymerase subunit), MCM3 [part of the MCM2-7 DNA helicase complex[39]], as well as the mismatch repair factor PMS2 that is a component of the MutLα complex[37]. Interestingly, MCM2, another member of the MCM2-7 complex has recently been implicated in CENP-A maintenance during S phase[45].

Proteins that we find uniquely involved in the CENP-A assembly process without any appreciable impact on CENP-A maintenance are the known dedicated CENP-A assembly factors Mis18α, Mis18β, MIS18BP1, and the CENP-A specific chaperone HJURP. Added to these we find CENP-R that was not previously implicated. Further we find ASF1B, a histone H3 chaperone involved in nucleosome recycling during DNA replication[39]. One hypothesis for its involvement in assembly may be that it acts as an acceptor protein for H3 exchange during CENP-A assembly. Further we find the chromatin modifiers, SMYD2 and NSD2[(WHSC1)], (both SET domain-containing proteins) and SUV39H2, a methyltransferase for histone H3 lysine 9 tri-methylation[73], critical for

heterochromatin function. The latter is of interest as heterochromatin has been implicated in CENP-A assembly and centromere formation[74,75]. The Polycomb proteins and histone methyl binding proteins CBX7 and L3MBTL[40,76] involved in transcription repression are also found to significantly affect assembly of new CENP-A. Finally, we found several proteins involved in ubiquitin metabolism, UBE2A (Ubiquitin Conjugating Enzyme E2 A), a Rad6 homolog involved in H2B ubiquitylation[77], KEAP1, an adapter protein for E3 ubiquitin ligase complexes and BRCC3, a BRCA1 and 2-associated Lys63-specific deubiquitinating enzyme[78], all affecting CENP-A loading.

Note that we discuss here the list of top candidates with an arbitrary cut-off of over 1.3 fold effect on CENP-A maintenance or 1.9 fold for CENP-A assembly. More genes were identified in the screen with a highly significant impact on CENP-A but with a lower fold difference (listed in Supplementary Data 2 and 3). It should also be noted that most candidates reported here, remain simply candidates as their role in CENP-A dynamics will require further confirmation and validation before their putative role in centromere biology can be established.

The candidate with the most significant effect on CENP-A maintenance, as well as assembly is the SUMO protease SENP6. No other SUMO regulators were found having such high impact on CENP-A inheritance. SENP6 is involved in removal of SUMO2/3 chains from target proteins[31]. Loss of SENP6, both long-term and acutely, results in the loss of both ancestral and nascent CENP-A from chromatin. In fact, we find a rapid disassembly of almost the entire centromere and kinetochore complex and SENP6 is required for CENP-A chromatin integrity at any point in the cell cycle. These findings suggest that the centromere complex is under continued surveillance by SUMO E3 ligases that control the localization of centromere proteins which is counteracted selectively by SENP6 (Supplementary Fig. 5).

Using SUMO pull-down experiments from cells in which we can control SENP6 levels, we identified CENP-C and CENP-I as targets for SENP6 control. Interestingly, four parallel studies came across a similar SENP6-CCAN pathway, from SUMO-mass spec analysis of proteins whose SUMOylation is modulated by SENP6[54], to analyzing the SENP6 interactome[55], to our work and that of ref.[38] that identified SENP6 by genetic screening for factors maintaining CENP-A chromatin.

In addition to CENP-C and CENP-I, both CENP-T and CENP-B were also found to be hyper-SUMOylated in absence of SENP6[54,55], although we did not find an effect on CENP-B localization in our system. These studies, as well as our work reported here did not detect SUMOylated species of CENP-A, indicating that CENP-A is not a direct target for SENP6. However, our studies did show that pre-incorporated CENP-A can be lost from the centromeres at any stage of the cell cycle upon depletion of SENP6. Taken together, these observations lead us to hypothesize that the primary role of SENP6 is to maintain the centromere complex in a hypo-SUMOylated state which may preserve protein-protein interactions and maintains the complex architecture of the CCAN. An intact CCAN, in turn, is required for protecting the centromeric CENP-A from eviction possibly by general chromatin disruptive processes such as replication, transcription or DNA repair (Supplementary Fig. 5). Indeed, CENP-C is known to stabilize CENP-A both in vitro and in vivo and is a central organizer of the centromere complex[9,79]. This protection would serve two important purposes for preserving centromere identity: (1) To maintain pre-assembled CENP-A at centromeres until the next G1 to act as a template for a new round of CENP-A loading and (2) to prevent ectopic kinetochore formation at other places in the chromosome. To fulfill the above two requirements SENP6 may be recruited and/or stabilized specifically at the kinetochore, stabilizing the local CCAN. Therefore future research

would focus on finding the targeting domains and the relevant binding partners of SENP6 in the inner kinetochore.

In sum, we report the identification of a series of proteins not previously associated with centromere structure and function. These factors act selectively in the assembly of new CENP-A, in the maintenance of chromatin-bound CENP-A or both and reveal the dynamic nature of the maintenance of centromeric chromatin. The identification of these factors serves as resource for further discovery into the control of centromere assembly and inheritance.

## Methods

**DNA constructs.** Constructs to build the SENP6$^{EGFP-AID/EGFP-AID}$ cell line are as follows: The plasmid pX330-U6-Chimeric_BB-CBh-hSpCas9 from Feng Zhang lab [Addgene #42230[80],] was used to construct the CRISPR/Cas vector plasmid according to the protocol in ref. [81]. Two guide RNA sequences: 5'-GCAAGAG CGGCGGTAGCGCA-3' (sg1) and 5'-GCCATGGATTAAGAAGGAGG-3' (sg2), designed to target the N terminal region of the SENP6 gene, were cloned into the pX330 backbone to generate the CRISPR/Cas vector plasmids pLJ869 (sg1) and pLJ870 (sg2), respectively. For generation of the N terminal AID tag, the construct LoxP-EGFP-LoxP-3xFLAG-miniAID-3xFLAG was gene synthesized and cloned into a pUc based vector to generate the template plasmid pLJ851. The homology donor vectors were constructed by PCR amplifying the template plasmid pLJ851 using Q5 DNA polymerase (New England Biolabs) with 110-base oligonucleotides using a 80-base homology sequence to the N terminal region of the SENP6 gene. The sequence of the upstream (US) and downstream (DS) homology arms are as follows: SENP6-US-HR-5'-CCGGCGCGGCCCCTCATCCCGGCGAGCACGGCG GCGGTGTGGGCCATGGATTAAGAAGGAGGCGGCGTGGGAGGAGGAAG' and SENP6-DS-HR-5'-GCGGCCGGCCAAGAGCGGCGGTAGCGCAGGGGAG ATTACTTTTCTGGAAGGTACGTCTGTTTCTGCCCTTGACGGGGAGAAGG GAG'. In both cases homology arms were designed to introduce silent mutations in the PAM (protospacer-adjacent motif) recognition sequence after integration into the target locus in order to prevent Cas9 re-cutting. The wild-type EGFP-SENP6 and the catalytic mutant EGFP-SENP6$^{C1030A}$ plasmids were gifts from Ronald Hay. The His10-SUMO2 plasmid was a gift from Alfred Vertegaal.

**Cell lines and culturing conditions.** All human cell lines were grown at 37 °C, 5% CO2. Cells were grown in DMEM (Bio West) supplemented with 10% new born calf serum (NCS) (Bio West) or 10% fetal bovine serum (FBS), 2 mM glutamine, 1 mM sodium pyruvate (SP) (Thermo Fisher Scientific), 100 U/ml penicillin and 100 μg/ml streptomycin. The SENP6$^{EGFP-AID/EGFP-AID}$ cell line as shown in Fig. 3a was constructed as follows: The parent cell line used was HeLa-CENP-A-SNAP clone #72[10,30]. This cell line was transduced with pBABE-OsTir1-9Myc retrovirus (pLJ820] gift from Andrew Holland, Johns Hopkins[49]] following the protocol in ref. [30]. The infected cells were selected by 500 μg/ml of Neomycin (Gibco). Individual resistant cells were sorted by FACS. In order to generate genome targeted cell line, a single clone expressing OsTIR1 was amplified and grown in 10 cm dishes before transfecting them with CRISPR/Cas vector plasmids (pLJ869 and pLJ870) and homology donor derived from pLJ851 using Lipofectamine LTX (Invitrogen; Carlsbad, CA) according to manufacturer's instructions. Monoclonal GFP positive clones were sorted by FACS. These clones were screened for homozygous tagging of the SENP6 gene by immunoblot using sheep anti-SENP6 antibody gift from Ronald Hay, Dundee. Based on the immunoblotting results a single clone (#18) was selected for performing auxin (Indole-3-acetic acid sodium salt or IAA; Sigma-Aldrich Cat. no. 15148) based experiments. Auxin (IAA) was used at concentration of 500 μM. For Auxin washout experiments, IAA-containing medium was aspirated, cells were washed twice in PBS and another two times in pre-warmed media followed by addition of fresh medium without auxin. For the His10-SUMO2 immuno-precipitation (IP) experiments, the EGFP-AID-SENP6 (HeLa-CENP-A-SNAP) line was transfected with His10-SUMO2 (Puro) plasmid gift from Alfred C. O Vertegaal) and selected by 1 μg/ml of Puromycin (Gibco). The puromycin resistant polyclonal population was used to perform His10-SUMO2 IP experiments. The primary cell line used was derived from healthy fetal skin fibroblast (Coriell Cell Repository #GM06170). All cell lines scored negative for Mycoplasma.

**siRNA library.** All siRNAs used in this study were obtained from Ambion Thermo Fisher Scientific as Silencer Select reagents. All siRNA sequences listed in Supplementary Data 1 were blasted for unique target specificity against the current human genome using ENSEMBL V95, 2019 (EMBL, EBI, and Wellcome Trust Sanger Institute).

**Transfection of siRNAs.** All siRNAs used in this study were purchased from the Silencer Select collection of siRNAs from Thermo Scientific and are listed in Supplementary Data 1. Production of RNAi microarrays is described in detail in ref. [82]. In brief, the siRNA-gelatin transfection solution was prepared in 384 well V-shaped plates using a manual 96 well pipetting device (Liquidator from Steinbrenner). For reverse solid transfection on cell arrays, 5 μl of 3 μM siRNA was mixed with 1.75 μl of Lipofectamine 2000 (ThermoFisher), 1.75 μl H2O and 3 μl of Opti-MEM (ThermoFisher) containing

0.4 M sucrose and incubated for 20 min at room temperature (per well protocol). Next, 7.25 μl of 0.2% gelatin (w/v) was added and the mixture was printed on one-chamber Lab-Tek slides (Nunc) with a contact printer (BioRad) eight solid pins, giving a spot size of ~400 μm diameter. The spot-to-spot distance was set to 1125 μm. Each Lab-Tek chamber accommodated 384 spots organized in 12 columns and 32 rows. The whole library of 2172 siRNAs was spotted onto seven Lab-Tek chambers, with 10 negative control siRNAs distributed randomly across each layout. After drying, the spotted library was seeded with CENP-A-SNAP pulse labeled cells as outlined in the section below.

For low throughput siRNA experiments, liquid-phase reverse transfection was performed on coverslips in 24 well plates using Lipofectamine RNAiMAX (Thermo Scientific) according to the manufacturer's protocol with $3 \times 10^4$ cells seeded per well. Cells were typically incubated for 48 h with the siRNA unless otherwise stated.

**SNAP pulse-chase and quench-chase-pulse labeling.** Cell lines expressing CENP-A-SNAP were pulse labeled as previously described[30]. For the primary siRNA screen, HeLa-CENP-A-SNAP cells were grown in 6 well plates and pulse labeled with tetra-methyl-rhodamine-conjugated SNAP substrate (TMR-Star; New England Biolabs) at 4 μM final concentration, labeling all pre-existing CENP-A molecules at the centromere. This was followed by a quenching step with bromothenylpteridine (BTP; New England Biolabs) at 2 μM final concentration to prevent any further fluorescent labeling of nascent CENP-A following TMR-Star washout. The cells were then trypsinized (Trypsin-EDTA, Gibco) and $1.5 \times 10^5$ cells were directly seeded onto 1 chamber Lab-Tek slides spotted with the siRNA library as described above. The chambers were then incubated at 37 °C, 5% CO2 for 48 h. At the end of 48 h, cells were labeled with Oregon-Green SNAP substrate (New England Biolabs) at 4 μM final concentration for labeling of the newly synthesized CENP-A molecules. Finally the cells were co-extracted (pre-extraction and fixation) in 4% paraformaldehyde, 0.2% Triton X and Hoechst (1 μg/ml) at room temperature for 30 min.

**Cell synchronization.** Double thymidine-based synchronization was performed as detailed in ref. [30]. For G2 synchronization, cells released from double thymidine arrest were incubated 4 h later with CDK1 inhibitor RO-3306 (Merck millipore) at 9 μM concentration for 4 h. For mitotic arrest and release cells were incubated with 2.5 μM of EG5 inhibitor Dimethylenastron for 13 h. Following inhibitor washout, cells were released for 9 h to obtain a synchronous population of late G1 cells.

**Immunofluorescence procedures.** The immunofluorescence procedures were based on ref. [30]. Briefly, the cells were grown on glass coverslips coated with poly-L lysine (Sigma-Aldrich) and fixed with 4% formaldehyde (Thermo Scientific) for 10 min followed by permeabilization in PBS with 0.1% Triton-X-100. When staining with antibodies for CENP-A, CENP-C, CENP-H, DSN1, NNF1, and HEC1, an additional pre-extraction step was included which involved incubation of cells with 0.1% Triton-X-100 in PBS for 5 min prior to fixation by formaldehyde. The following antibodies and dilutions were used: mouse monoclonal anti-CENP-A [gift from Kinya Yoda (Nagoya University)[83]] at 1:100; rabbit polyclonal anti-CENP-B (sc22788; Santa Cruz Biotechnology, Dallas, TX) at 1:1000; rabbit polyclonal anti-CENP-B (ab25734; Abcam) at 1:500; mouse monoclonal anti-CENP-B (ab167361; Abcam) at 1:100; mouse monoclonal anti-CENP-C isolated from hybridoma line (LX191) [gift from Don Cleveland, UCSD[23]] at 1:100, rabbit polyclonal anti CENP-T [gift from Don Cleveland, UCSD[5]; rat monoclonal anti-CENP-H and rabbit polyclonal anti-CENP-I (both gifts from Song-Tao Liu, University of Toledo); rabbit polyclonal anti-DSN1 (gift from Iain Cheeseman, Whitehead) at 1:100; rabbit polyclonal anti-NNF1 (gift from Arshad Desai, UCSD); mouse monoclonal anti-HEC1 (Thermo Scientific Pierce MA1-23308), sheep polyclonal anti-SENP6 gift from Ronald Hay, Dundee) at 1:500; rat monoclonal anti-Tubulin (SC-53029, Santa Cruz Biotechnology, Dallas, TX) at 1:10,000 and mouse monoclonal anti Cyclin-B (SC-245, Santa Cruz Biotechnology, Dallas, TX) at 1:50. All primary antibody incubations were performed at 37 °C for 1 h in a humid chamber. Fluorescent secondary antibodies were obtained from Jackson ImmunoResearch (West Grove, PA) or Rockland ImmunoChemicals and used at a dilution of 1:200. All secondary antibody incubations were performed at 37 °C for 45 min in a humid chamber. Cells were stained with DAPI (4',6-diamidino-2-phenylindole; Sigma-Aldrich) before mounting in Mowiol. EdU (5-ethynyl-2'-deoxyuridine) labeling was performed for 15 min as per the manufacturer's instructions (C10340, Life Technologies) in order to stain S phase nuclei. In the experiments where EdU labeling was performed, EGFP-SENP6 signal was detected using GFP-Booster Atto488 (Chromotek).

Immunofluorescent signals of Figs. 2–5, Supplementary Figs. 3 and 4 were quantified using the CRaQ (Centromere Recognition and Quantification) method[30] using CENP-B as centromeric reference. Hec1, Dsn1, and Nnf1 levels were measured only in prometaphase or metaphase (based on DAPI staining) nuclei. The immunofluorescent signal of EGFP-SENP6 in Supplementary Fig. 2 was measured using an ImageJ based macro which measured the median intensity of the whole nucleus.

**Microscopy.** Imaging for the primary siRNA screen was performed on an Olympus ScanR (IX-81) automated inverted microscope (Olympus Biosystems),

equipped with a Hammamatsu Orca-ER and a MT20 light source. The microscope was controlled by ScanR acquisition software (version 2.3.0.7). A ×20 0.75 NA air objective (UpLANsaPO; Olympus Biosystems) was used for the primary screening on cell arrays and a single plane image was acquired for each siRNA spot. Filter settings and exposure times were the following: Dapi: Ex: 347/50 EM: 460/50, exposure time 20 ms; Oregon Green Ex: 482/35 EM: 536/40, exposure time 500 ms and TMR star: 545/30 EM: 610/72 exposure time 1500 ms.

For validation and characterization experiments imaging was performed using either of the two following systems: (a) a Deltavision Core system (Applied Precision) inverted microscope (Olympus, IX-71) coupled to Cascade2 EMCCD camera (Photometrics). Images (512 × 512) were acquired at 1× binning using a ×100 oil objective (NA 1.40, UPlanSApo) or a ×60 oil objective (NA 1.42 PlanApoN) with 0.2 μm z sections. (b) Leica High Content Screening microscope, based on Leica DMI6000 equipped with a Hamamatsu Flash Orca 4.0 sCMOS camera, using a ×100 1.44 NA objective (HC PLAN APO) or a ×63 1.4 NA objective (HC PLAN APO) with 0.2 μm z sections

**Image analysis**. Image analysis of the primary screen was performed using a Cell-Profiler[84] pipeline, available for download (https://git.embl.de/grp-almf/sreyoshi-mitra-jensen-centromere-screen/blob/master/publication/mitra_centromere_screen_cp2.2.0.cpproj) and can be used with CellProfiler version 2.2.0. In brief, nuclei were detected in the DAPI image using automated thresholding. Based on their texture, non-interphase (mitotic or other, e.g., dead cells) nuclei were removed from the analysis. The 'old' and 'new' CENP-A-SNAP images were subjected to a morphological tophat filter in order to remove diffuse (non-centromeric) background signal. Centromeric regions were detected using automated thresholding and, for each nucleus, the integrated intensity within centromeric regions was measured. For downstream data analysis, we computed the mean values of all nuclei in each image, yielding, for each image, two measurements: "Mean_Interphase Nuclei_Intensity_Integrated Intensity_New Tophat Centromere Mask" and "Mean_Interphase Nuclei_Intensity_Integrated Intensity_Old Tophat Centromere Mask". In addition, to be able to reject out-of-focus images, we used CellProfiler's Measure Image Quality module, specifically the values: Image-Quality_Power Log Log Slope_Nucleus and ImageQuality_Power Log Log Slope_Old. Those are spatial frequency based measurements, where low values indicate missing high spatial frequencies such as it is the case for out-of-focus images.

**Statistical analysis**. Visual inspection, quality control and statistical analysis of the primary screen was performed using HTM explorer. Primary data was the CellProfiler output table, containing measurements of 13440 images, corresponding to 35 384 spotted Lab-Tek chambers. In terms of quality control, we filtered out-of-focus images, rejecting all images where either ImageQuality_PowerLogLogSlope_Nucleus <−2.3 or ImageQuality_PowerLogLogSlope_Old <−2.0, thereby removing 596 images from the analysis. The threshold values −2.3 and −2.0 were determined by visual inspection. The aim of the primary screen was to measure the siRNA knockdown induced fold-change of CENP-A centromere signal relative to our negative control siRNA. To this end, we first computed a $\log_2$ transform of both CENP-A readouts (Mean_InterphaseNuclei_Intensity_IntegratedIntensity_NewTophatCentromere Mask and Mean_InterphaseNuclei_Intensity_IntegratedIntensity_OldTophat CentromereMask). Next, for both readouts, we performed a Lab-Tek chamber-wise normalization by subtracting, for each chamber, the mean value of the 10 negative control images on that plate. To obtain one final score per treatment (siRNA) we pooled the normalized values for each treatment across all plates and performed a t-test against the normalized negative control values from each plate. The t-test's estimate of the difference between treatment and control represents the $\log_2$ fold ratio of treatment and control. The t-test's p-value represents the statistical significance of this difference.

**Parameters as listed in the Supplementary Data 2 and 3**. Fold difference vs. control (t-test position estimate): In the first step, data to control measurements from a single chamber is normalized by subtracting the mean of the control positions in that chamber. Afterwards, for one specific treatment all chambers are identified that contained this treatment and all treatment and control measurements from these batches are pooled. The t test positions estimate is the difference of treated positions and control positions (after batch correction). If the $\log_2$ data transformation of this value is chosen, this difference gives the fold-change of treatment vs. control (in $\log_2$ scale). To compute the actual fold change you can use this formula: $2^{estimate}$.

Significance (t-test p value): After performing batch correction, for one specific treatment all chambers are identified that contained this treatment and all treatment and control measurements from these batches are pooled; a t-test is performed of the treated positions against the control positions. t test positions p value gives the p value computed from the above t test.

Median z score: For each chamber, the data of all images within one position are averaged such that we have one number per position. Then a z-score is computed for each position as $Z = (value − mean (ctrls))/sd (ctrls)$ Where the mean and standard deviation (sd) are computed across all positions that contain the selected control measurements. Median z-score is the median value of z-scores from multiple chambers containing the specific treatment.

Median robust z score: Same as median z score, but for each chamber a z-score is computed as $Z = (treated − median (ctrls))/mad (ctrls)$, where mad is the so called median average deviation (a median based analog to the standard deviation).

To represent the final dataset in Fig. 1, the Fold difference vs. control [t test position estimate values (equivalent to $\log_2$ fold change of siRNA treatment vs. control)] and the negative $\log_{10}$ p-values were plotted as volcano plots using GraphPad Prism version 7. Two volcano plots were generated depicting the candidates affecting the loading of new CENP-A molecules and those affecting the maintenance of pre-assembled old CENP-A at the centromere, hereafter called 'loading' and 'maintenance' candidates, respectively. For the maintenance candidates, a cut-off of −0.4 was set for $\log_2$ fold-change representing at least 1.3-fold reduction in old CENP-A intensity in the siRNA treated condition vs. that in a negative scrambled siRNA control. For the loading candidates, a cut-off of −1 was set for $\log_2$ fold-change representing at least 2-fold reduction in new CENP-A intensity in the siRNA treated condition vs. that in a negative scrambled siRNA control. A cut-off value of 3 for –log (p-value), corresponding to a p-value of <0.001, was employed to ascribe statistical significance for both maintenance and loading candidates.

**His10-SUMO2 purification**. His10-SUMO2 conjugates were purified as described[54]. Briefly, EGFP-AID-SENP6 (HeLa-CENP-A-SNAP) cells expressing His10-SUMO2 under selection, were lysed in 25 volumes of 6 M Guanidium-hydrochloride, 0.1 M Na$_2$HPO$_4$/NaH$_2$PO$_4$, 0.01 M Tris-HCl, pH 8.0. The lysed cells were briefly sonicated using a QSonica bath sonicator for 2 × 5 s at 50% amplitude and then 50 mM imidazole (pH 8) and 5 mM β-mercaptoethanol was added to the lysates. Meanwhile Ni-NTA beads (Qiagen, 36111) were pre-washed with Lysis Buffer supplemented with 50 mM imidazole pH 8 and 5 mM β-mercaptoethanol. The washed beads were incubated with the lysates overnight at 4 °C. Next day, the beads were washed with Wash buffers 1-4 (Wash Buffer 1: 6 M guanidine-HCl, 0.1 M Na$_2$HPO$_4$/NaH$_2$PO$_4$ pH 8.0, 0.01 M Tris-HCl pH 8.0, 10 mM imidazole pH 8.0, 5 mM β-mercaptoethanol, 0.2% Triton X-100. Wash Buffer 2: 8 M urea, 0.1 M Na$_2$HPO$_4$/NaH$_2$PO$_4$ pH 8.0, 0.01 M Tris-HCl pH 8.0, 10 mM imidazole pH 8.0, 5 mM β-mercaptoethanol, 0.2% Triton X-100. Wash Buffer 3: 8 M urea, 0.1 M Na$_2$HPO$_4$/NaH$_2$PO$_4$ pH 6.3, 0.01 M Tris-HCl pH 6.3, 10 mM imidazole pH 7.0, 5 mM β-mercaptoethanol, 0.2% Triton X-100. Wash Buffer 4: 8 M urea, 0.1 M Na$_2$HPO$_4$/NaH$_2$PO$_4$ pH 6.3, 0.01 M Tris-HCl pH 6.3, no imidazole, 5 mM β-mercaptoethanol, 0.1% Triton X-100.) Subsequently, the conjugated proteins were eluted twice in one bead volume of elution buffer (7 M urea, 0.1 M Na$_2$HPO$_4$/NaH$_2$PO$_4$, 0.01 M Tris-HCl, pH 7.0, 500 mM imidazole pH 7.0).

**Immunoblotting**. Whole cell extracts were prepared by direct lysis in 1× Laemmli sample buffer, separated by SDS-PAGE and transferred onto nitrocellulose membranes. The following antibodies and dilutions were used: rabbit polyclonal anti-CENP-A (#2186, Cell Signaling Technology) at 1:500; mouse monoclonal anti-α tubulin (T9026, Sigma-Aldrich) at 1:5000; rabbit polyclonal anti-CENP-C (Covance) gift from Don Cleveland, UCSD) crude serum at 1:5000, sheep polyclonal anti-SENP6 (gift from Ronald Hay, Dundee) at 1:5000, mouse monoclonal anti-SUMO2 (Abcam, ab81371), rabbit polyclonal anti-CENP-I at 1:500 gift from Song-Tao Liu, University of Toledo) and rabbit polyclonal anti-MIS18BP1 at 1:1000 (Bethyl, A302-825A). IRDye800CW-coupled anti-rabbit (Licor Biosciences), DyLight800-coupled anti-rabbit (Rockland Immunochemicals), DyLight800-coupled anti-mouse (Rockland Immunochemicals) and DyLight680-coupled anti-mouse (Rockland Immunochemicals) secondary antibodies were used at 1:10,000 prior to detection on an Odyssey near-infrared scanner (Licor Biosciences).

**Reporting summary**. Further information on research design is available in the Nature Research Reporting Summary linked to this article.

## Data availability

All relevant data supporting the key findings of this study are available within the article and its Supplementary Information files or from the corresponding author upon reasonable request. The source data for Figs. 2–6 and Supplementary Figs. 2, 3, 4 are provided as a Source Data files 1 and 2, which includes all image quantification data and raw immunoblots, respectively. A reporting summary for this article is available as a Supplementary Information file.

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

## Acknowledgements

We thank Andrew Holland (Hopkins), Feng Zhang (MIT), Ronald Hay (Dundee), Song-Tao Liu (University of Toledo), Alfred Vertegaal (LUMC), Ian Cheeseman (Whitehead), Ben Black (UPenn) Arshad Desai, and Don Cleveland (both UCSD) for reagents. We thank Sebastiaan van den Berg for critically reading the paper. This work is supported by an ERC-consolidator grant ERC-2013-CoG-615638 and a Senior Wellcome Research Fellowship, both to LETJ. Further salary support to D.L.B. was provided by Fundação para a Ciência a e Tecnologia (FCT) fellowship SFRH/BD/74284/2010 and a Sir Henry Wellcome Fellowship [204747/Z/16/Z] and an "Investigador FCT" position to LETJ.

## Author contributions

S.M. designed and performed the experiments, designed figures, and wrote the paper. D.L.B. and A.F.D. performed initial pilot siRNA screen. I.A.Z. performed experiments in primary cells. J.M. provided tissue culture expertize and assisted in cell line construction. B.N. and S.R. provided essential guidance to high through put siRNA screening. C.T. built the data analysis pipeline. L.E.T. conceived the study, helped design the experiments, wrote the paper and acquired funding.

## Competing interests

The authors declare no competing interests.
