## [Peer Review File · Nature Communications]

Reviewers' comments:

Reviewer #1 (Remarks to the Author):

Centromere are composed of a complex set of proteins built on a unique centromere chromatin enriched in the histone H3 variant CENP-A. The stability and the replenishment of new CENP-A at every cell cycle is key for centromere function. Despite many great works described how stability and replenishment are achieved, other unknown factors can be involved.

In this manuscript, Mira et al. performed a genetic screen based on a 2172 siRNAs library (encompassing 1046 genes) coupled with microscopy analysis to identify candidate genes affecting centromere CENP-A stability and assembly. By using the SNAP-tag pulse chase labelling strategy the authors could distinguish "old" CENP-A from the "newly" deposited CENP-A molecules. Among many interesting factors, the screen revealed that the SUMO-specific protease SENP6 was particularly important for CENP-A stability. The second part of the study focuses on further characterizing SENP6's role in the maintenance of centromere and kinetochore proteins (mainly on CENP-A), using an auxin inducible degron tag.

Overall, the experiments described in the paper are well designed, performed and convincing. The genetic screen performed in this study constitutes a very compelling and useful database for the centromere research field. There are, however, few points that should be addressed before publication in Nature Communications.

Major points

1. One of the big surprises reading this manuscript is that SENP6 depletion affects both CENP-A and CENP-C localization at the centromere but doesn't lead to protein degradation. CENP-A was reported to be SUMOylated in budding yeast (Ohkuni et al 2016, 2018), but CENP-A SUMOylation-regulated proteosomal degradation has not been shown yet in human. The authors should provide a better explanation/mechanism with susceptible targets of SENP6 that could directly or not impact CENP-A maintenance (without leading to its degradation) or propose intermediate partners. For instance, is loss of SENP6 increasing histone turnover, which could explain the loss of CENP-A nucleosomes? An immunoprecipitation experiment for CENP-A and CENP-C under SENP6 auxin depletion control could also be performed to identify potential candidates or modifications that control their association to the centromeric regions.
2. As CENP-A and CENP-C protein levels are not affected, is the loading of CENP-A reversible after SENP6 re-expression? The authors could perform an auxin washout experiment or ectopically overexpress SENP6 to rescue the phenotype.
3. As CENP-A and CENP-C are key factors to maintain all the centromere and kinetochore network, it will be interesting to test if the effect that the authors see on other centromeric proteins is due to CENP-A (or CENP-C) loss. This leads to another key point: there is no data showing that, in the absence of SENP6, CENP-A-SUMO accumulates in the cells. The authors should test this, and they could do the same for other centromeric proteins, so they will also be able to answer my previous question (direct or indirect effect of centromeric protein loss due to CENP-A removal).
4. Figure 4: The authors performed a long cell synchronization assay but they do not provide data showing that the cells are arrested at the various time points. They could for example present the different cell cycle arrests by flow cytometry analysis. Also, it is not clear if and how SENP6 depletion affects the cell cycle.
5. Figure 4: It is unclear if CENP-A stability was measured using the SNAP tag. While this is made clear in the cell cycle related experiments, there is no information if the authors use TMR-SNAP to measure CENP-A or, more logically, a CENP-A antibody. Also, why measure CENP-A transgene levels in the SENP6-AID cell line and not simply endogenous CENP-A?

Minor points

1. Figure 3 D: Dsn1 and CENP-B staining in siSENP6 condition doesn't reflect the signal

- quantification in comparison with siCtrl. The authors should provide a better example of a metaphase cell or enlarge their IF images.
2. No statistical analyses are provided concerning all the normalized quantifications of signal intensity.
 3. The authors should have made clearer that their screening cannot really distinguish between maintenance and de novo loading if the stability of the protein itself is rapidly affected.
 4. Line 131: different font/size.
 5. Line 198: As SENP6 signal is still present, according to their quantification (about half), the authors should rephrase to tone down their statement (e.g. rapid reduction, partially undetectable, ...)
 6. Line 212: it should be 6hr accordingly to their schematic.

Reviewer #2 (Remarks to the Author):

In this study, Mitra et al use a microscopy-based screening approach to identify regulators of centromere assembly and maintenance. They use an elegant timer approach based on pulse-labelling of a SNAP-tagged centromere subunit, CENP-A, to determine the lifetime of the protein within the centromere as well as its new assembly in time. With this approach they screen a collection of siRNAs representing more than 1000 genes for effects on either maintenance or new assembly of CENP-A. Their strongest hit, the SUMO-isopeptidase SENP6, is subjected to an in-depth characterisation, and the authors find that the protein is required continuously to prevent acute loss of CENP-A from chromatin. They ascribe this to a cell-cycle-independent effect, as it is observable in all cell cycle phases.

The manuscript is nicely written and the experiments are well designed and clean. In some ways, it is a hybrid between a screen and a mechanistic study. The tagging approaches are elegant, as the SNAP-tagging allows a precise observation of the age and lifetime of proteins, while the degron-tagging gives valuable information about short-term effects of SENP6 all along the cell cycle. It is important to use methods like these to gain insight into the mechanism of protein function, as they go far beyond the standard siRNA or even CRISPR/Cas approaches in their precision. In that sense, the manuscript is a valuable showcase of how a sub-cellular structure can be examined in a time-resolved manner.

The strength of the manuscript is clearly in these detailed mechanistic observations, while the overall novelty is more limited. A contribution of SENP6 to centromere assembly has been described before, but the novelty of the authors' findings lies in their observation that the isopeptidase also regulates maintenance of the centromere. The effects of SENP6 depletion are dramatic and fast, as the authors show by means of a degron-tagged SENP6. As CENP-A is one of the innermost components of the centromere, its loss from chromatin causes a substantial collapse of the centromere and kinetochore structure with widespread loss of other components. Thus, the effect of SENP6 is striking and might explain in parts the essential nature of the protein.

Other candidates resulting from the screen are listed and discussed in rather great length, but without a careful validation. Thus, while the screening technique itself is impressive, the actual results are somewhat preliminary and have to be viewed with caution.

I have very few comments as to what should be done to strengthen the manuscript. The one obvious question that is not answered by the study concerns the possible substrates of SENP6 that are responsible for the phenotype. The authors observe SENP6 in proximity to the centromeres, but they don't show that the relevant substrates are in fact - as one would expect - centromere

components. I appreciate the fact that they do not over-interpret their results and make claims about possible substrates, but the manuscript might benefit from examination of some candidates, such as CENP-C, M18BP1 or CENP-A itself. They could also try to suppress the effects of SENP6 loss by fusion of a non-specific SUMO isopeptidase domain to one of the centromere components in order to test the prediction that the centromere itself is the relevant target.

REVIEWER #1:

Centromere are composed of a complex set of proteins built on a unique centromere chromatin enriched in the histone H3 variant CENP-A. The stability and the replenishment of new CENP-A at every cell cycle is key for centromere function. Despite many great works described how stability and replenishment are achieved, other unknown factors can be involved.

In this manuscript, Mira et al. performed a genetic screen based on a 2172 siRNAs library (encompassing 1046 genes) coupled with microscopy analysis to identify candidate genes affecting centromere CENP-A stability and assembly. By using the SNAP-tag pulse chase labelling strategy the authors could distinguish “old” CENP-A from the “newly” deposited CENP-A molecules. Among many interesting factors, the screen revealed that the SUMO-specific protease SENP6 was particularly important for CENP-A stability. The second part of the study focuses on further characterizing SENP6’s role in the maintenance of centromere and kinetochore proteins (mainly on CENP-A), using an auxin inducible degron tag.

Overall, the experiments described in the paper are well designed, performed and convincing. The genetic screen performed in this study constitutes a very compelling and useful database for the centromere research field. There are, however, few points that should be addressed before publication in Nature Communications.

Major points

1. One of the big surprises reading this manuscript is that SENP6 depletion affects both CENP-A and CENP-C localization at the centromere but doesn’t lead to protein degradation. CENP-A was reported to be SUMOylated in budding yeast (Ohkuni et al 2016, 2018), but CENP-A SUMOylation-regulated proteosomal degradation has not been shown yet in human. The authors should provide a better explanation/mechanism with susceptible targets of SENP6 that could directly or not impact CENP-A maintenance (without leading to its degradation) or propose intermediate partners. For instance, is loss of SENP6 increasing histone turnover, which could explain the loss of CENP-A nucleosomes? An immunoprecipitation experiment for CENP-A and CENP-C under SENP6 auxin depletion control could also be performed to identify potential candidates or modifications that control their association to the centromeric regions.

The reviewer identifies several important points that follow from our work on SENP6. We feel the latter question to be the most central one, i.e. what are the putative SUMO targets at the centromere? We have made a significant effort to address this point and have developed pull down assays from our SENP6 degron lines. We performed high affinity, stringent pull downs of His-tagged SUMO on nickel resin and probed these for CCAN components for which we could generate suitable reagents in a timely manner. We find significant CENP-C and CENP-I SUMOylation and to a lesser extent, MIS18BP1 (new Fig. 6). Perhaps unexpectedly, we find no evidence for CENP-A SUMOylation. Importantly, CENP-C and CENP-I SUMOylation is enhanced in SENP6 depleted cells indicating these proteins are bona fide SENP6 targets. While we have yet to discover how CENP-A is physically removed from the centromere (the reviewer’s first point), we believe that our new findings provide a tangible mechanistic basis for the SENP6 phenotype as previously loss of CENP-C was shown to lead to loss of CENP-A from chromatin.

It should be pointed out that during the revision, two additional studies were published that corroborate the general finding that SENP6 regulates CCAN localization via SUMOylation of multiple centromere proteins in a manner that does not lead to their degradation.

2. As CENP-A and CENP-C protein levels are not affected, is the loading of CENP-A reversible after SENP6 re-expression? The authors could perform an auxin washout experiment or ectopically overexpress SENP6 to rescue the phenotype.

The reviewer raises an interesting point and we took the reviewers' advice to test this in an auxin washout assay combined with SNAP-based CENP-A assembly. We find no evidence for the re-loading of the previously chromatin bound pool that was lost due to SENP6 depletion (new Fig. 4G, H). This suggests that while CENP-A released from chromatin is not degraded, it is no longer available in a form suitable for loading. In contrast, we find that nascent CENP-A loading can occur following auxin washout (new Fig. 4E, F). This indicates that while CENP-A, CENP-C and CENP-T levels are significantly reduced at centromere, the assembly machinery is still able to recognize the centromere for nascent CENP-A incorporation, as expected. Furthermore, our data suggests that centromeres can recover from excess SUMOylation.

3. As CENP-A and CENP-C are key factors to maintain all the centromere and kinetochore network, it will be interesting to test if the effect that the authors see on other centromeric proteins is due to CENP-A (or CENP-C) loss. This leads to another key point: there is no data showing that, in the absence of SENP6, CENP-A-SUMO accumulates in the cells. The authors should test this, and they could do the same for other centromeric proteins, so they will also be able to answer my previous question (direct or indirect effect of centromeric protein loss due to CENP-A removal).

We agree with the reviewer that this is a valid point and the issue of whether CENP-A stability is mediated by CENP-C is important. Based on our new SUMO pulldown experiments (described above, new Fig. 6), we observe no direct SUMOylation of CENP-A and conclude that the loss of CENP-A is likely an indirect effect of the hyper-SUMOylation of other CCAN components, possibly CENP-C.

4. Figure 4: The authors performed a long cell synchronization assay but they do not provide data showing that the cells are arrested at the various time points. They could for example present the different cell cycle arrests by flow cytometry analysis. Also, it is not clear if and how SENP6 depletion affects the cell cycle.

While we have not directly assessed whether SENP6 affects the cell cycle, a study contemporaneous to ours, recently published in this journal (Liebelt et al., Nature Comm. 2019), reported a delay in G2/M phases as measured by FACS. In our synchronization experiments, we analysed the cyclin B and EdU staining, marking G2 and S phase populations, to ensure that cells are enriched in the intended cell cycle phase. We have included the numbers for this in lines 192-194 of the manuscript.

Furthermore, we corroborated our cell synchronization data with experiments in cells that remained unperturbed but where cell cycle stages were identified with specific

markers (Fig. S3). These data confirm that SENP6 acts to maintain centromere integrity in G1, S and G2 phases of the cell cycle.

5. Figure 4: It is unclear if CENP-A stability was measured using the SNAP tag. While this is made clear in the cell cycle related experiments, there is no information if the authors use TMR-SNAP to measure CENP-A or, more logically, a CENP-A antibody. Also, why measure CENP-A transgene levels in the SENP6-AID cell line and not simply endogenous CENP-A?

The reviewer is correct that this was not clearly stated. Also, we have now performed additional experiments to address this point. To determine whether the effect of SENP6 depletion on CENP-A is not SNAP-tag specific nor cell type specific we have performed siRNA experiments both in parental HeLa cells (expressing only endogenous untagged CENP-A) as well as in primary human embryonic fibroblasts (Figure 2F-I). In both cases SENP6 results in loss of centromeric CENP-A as measured with a CENP-A antibody, corroborating our data in SNAP-tagged cells.

Minor points

1. Figure 3 D: Dsn1 and CENP-B staining in siSENP6 condition doesn't reflect the signal quantification in comparison with siCtrl. The authors should provide a better example of a metaphase cell or enlarge their IF images.

Thanks for pointing this out. We have selected a better example that is more representative of the average effect on Dsn1

2. No statistical analyses are provided concerning all the normalized quantifications of signal intensity.

Thank you. We have now fixed this oversight. We have included averages, variance of replicate experiments and relevant statistical tests for all quantifications throughout the manuscript.

3. The authors should have made clearer that their screening cannot really distinguish between maintenance and de novo loading if the stability of the protein itself is rapidly affected.

It is indeed the case that if a factor is involved in stabilizing/maintaining CENP-A chromatin, it may do so both for ancestral as well as newly incorporated CENP-A. As such we cannot determine whether a “maintenance” factor also has an additional role in loading. This would not withstand the fact that such a factor has a role in maintenance, although perhaps not exclusively. We hope we correctly understood the reviewers point. We have now added a statement reflecting this in lines: 123-125.

4. Line 131: different font/size.

Thank you, this is now fixed

5. Line 198: As SENP6 signal is still present, according to their quantification (about half), the authors should rephrase to tone down their statement (e.g. rapid reduction, partially undetectable, ...)

Based on our western blot data in Figure S2A as well as the fact that nuclear staining of SENP6 disappears following IAA addition (only cytoplasmic background is visible in most cells in Figure S2D), we don't believe there is half of the SENP6 pool still present following IAA addition. It is indeed the case that our graph in Figure S2B gives this impression as we use non-background-subtracted raw intensity values. However, based on the data mentioned above, it is reasonable to assume that by 3 hours of IAA most of SENP6 is depleted (and no further loss is observed by 48 hours). We have now added additional statements regarding this background level in the figure legend. We hope this clarifies this point.

6. Line 212: it should be 6hr accordingly to their schematic.

Thanks, this is now fixed.

REVIEWER #2

In this study, Mitra et al use a microscopy-based screening approach to identify regulators of centromere assembly and maintenance. They use an elegant timer approach based on pulse-labelling of a SNAP-tagged centromere subunit, CENP-A, to determine the lifetime of the protein within the centromere as well as its new assembly in time. With this approach they screen a collection of siRNAs representing more than 1000 genes for effects on either maintenance or new assembly of CENP-A. Their strongest hit, the SUMO-isopeptidase SENP6, is subjected to an in-depth characterisation, and the authors find that the protein is required continuously to prevent acute loss of CENP-A from chromatin. They ascribe this to a cell-cycle-independent effect, as it is observable in all cell cycle phases.

The manuscript is nicely written and the experiments are well designed and clean. In some ways, it is a hybrid between a screen and a mechanistic study. The tagging approaches are elegant, as the SNAP-tagging allows a precise observation of the age and lifetime of proteins, while the degraon-tagging gives valuable information about short-term effects of SENP6 all along the cell cycle. It is important to use methods like these to gain insight into the mechanism of protein function, as they go far beyond the standard siRNA or even CRISPR/Cas approaches in their precision. In that sense, the manuscript is a valuable showcase of how a sub-cellular structure can be examined in a time-resolved manner.

The strength of the manuscript is clearly in these detailed mechanistic observations, while the overall novelty is more limited. A contribution of SENP6 to centromere assembly has been described before, but the novelty of the authors' findings lies in their observation that the isopeptidase also regulates maintenance of the centromere. The effects of SENP6 depletion are dramatic and fast, as the authors show by means of a degraon-tagged SENP6. As CENP-A is one of the innermost components of the centromere, its loss from chromatin causes a substantial collapse of the centromere and kinetochore structure with widespread loss of other components. Thus, the effect of SENP6 is striking and might explain in parts the essential nature of the protein.

We appreciate the reviewer's positive comments. Indeed, SENP6 has been implicated in centromere biology before, initially by Mary Dasso's work on CENP-H and I (JCB 2010). We believe our work (along with recent other papers) now reveal that the role of

SENP6 is much more widespread at the centromere including a direct impact on centromeric chromatin through the cell cycle which is novel.

Other candidates resulting from the screen are listed and discussed in rather great length, but without a careful validation. Thus, while the screening technique itself is impressive, the actual results are somewhat preliminary and have to be viewed with caution.

We agree with the reviewer that the other candidates from our screen should be interpreted with caution as most of these have not yet been validated. This is generally the case for most screens and we offer this to the community as a resource for further study, not as established facts. We have carefully pointed this out now in line 375-377 of the manuscript.

I have very few comments as to what should be done to strengthen the manuscript. The one obvious question that is not answered by the study concerns the possible substrates of SENP6 that are responsible for the phenotype. The authors observe SENP6 in proximity to the centromeres, but they don't show that the relevant substrates are in fact - as one would expect - centromere components. I appreciate the fact that they do not over-interpret their results and make claims about possible substrates, but the manuscript might benefit from examination of some candidates, such as CENP-C, M18BP1 or CENP-A itself.

The reviewer raises an important point and we have made significant efforts to address this. We have now included new experiments in which we pull down His-tagged SUMO from HeLa cell extract derived from cells in which SENP6 was left intact or depleted (as also outlined in our response to reviewer 1). These are analysed for the presence of SUMOylated targets of those proteins for which we have good reagents available. We find evidence for SUMOylation of CENP-C, CENP-I, but not for CENP-A itself or M18BP1. This new data is also in line with contemporaneous studies, published during the revision of our work, that also find these proteins to be directly SUMOylated.

They could also try to suppress the effects of SENP6 loss by fusion of a non-specific SUMO isopeptidase domain to one of the centromere components in order to test the prediction that the centromere itself is the relevant target.

While we agree with the reviewer that this experiments would in itself be insightful, we and others have now identified several SUMO targets within the centromere complex that we find to be controlled by SENP6. These novel findings make this latter proposed experiment perhaps less relevant to our study.

REVIEWERS' COMMENTS:

Reviewer #1 (Remarks to the Author):

In the revised version of this paper, Mitra et al addressed several key points raised by the reviewers. This includes two important aspects: the SUMOylating state of several centromeric proteins and the reversibility of the SUMOylating process in term of centromeric protein stability. Despite the authors could not detect any direct CENP-A SUMOylation that still raised some open questions on how CENP-A is removed from centromeres (this is unlikely to be explained with the sole effect of CENP-C loss), the authors provided new strong experimental evidences to support their findings.

Overall, this is a very nice and well performed work that will be of great interest for the readers of Nature Communications and the chromatin community.

Minor points to consider

1. Line 246 to 249: the authors should consider to change the word "lost" to something like "reduced" when referring to the KMN protein. Indeed, they observed a significant decrease after IAA but the total signal is still more than half (>0.5).
2. The authors never measured endogenous CENP-A level in SENP6-AID cells but only in RNAi condition. This reviewer does not understand if there is any unexplained reason for that.
3. In the experimental section is missing the procedure for auxin washout

Reviewer #2 (Remarks to the Author):

The authors have carefully addressed the reviewers' comments and have provided additional support by a set of new experiments. I am now happy with the manuscript in its present shape and recommend publication.

REVIEWER #1

In the revised version of this paper, Mitra et al addressed several key points raised by the reviewers. This includes two important aspects: the SUMOylating state of several centromeric proteins and the reversibility of the SUMOylating process in term of centromeric protein stability.

Despite the authors could not detect any direct CENP-A SUMOylation that still raised some open questions on how CENP-A is removed from centromeres (this is unlikely to be explained with the sole effect of CENP-C loss), the authors provided new strong experimental evidences to support their findings.

Overall, this is a very nice and well performed work that will be of great interest for the readers of Nature Communications and the chromatin community.

Minor points to consider

1. Line 246 to 249: the authors should consider to change the word “lost” to something like “reduced” when referring to the KMN protein. Indeed, they observed a significant decrease after IAA but the total signal is still more than half (>0.5).

We have modified the text to reflect this (new line 219)

2. The authors never measured endogenous CENP-A level in SENP6-AID cells but only in RNAi condition. This reviewer does not understand if there is any unexplained reason for that.

The reason for this is technical. We generated the SENP6-AID alleles specifically to determine CENP-A maintenance by SNAP pulse/chase labelling in line with our original screen. For this reason we build this line in cells already expressing CENP-A-SNAP. Therefore these cells express both endogenous and SNAP-tagged CENP-A.

In order to assess truly endogenous pools we used parental unmodified HeLa cells (as well as primary cells) to look at the impact of SENP6 depletion, which in this context we could only perform by RNAi. We hope this clarifies this point.

3. In the experimental section is missing the procedure for auxin washout

We have now added the procedure for the auxin washout experiments (new lines 416-419)

REVIEWER #2

The authors have carefully addressed the reviewers' comments and have provided additional support by a set of new experiments. I am now happy with the manuscript in its present shape and recommend publication.